



# ESA CCI Soil Moisture GAPFILLED: An independent global gap-free satellite climate data record with uncertainty estimates

Wolfgang Preimesberger[1], Pietro Stradiotti[1], and Wouter Dorigo[1]

[1]Department of Geodesy and Geoinformation, TU Wien, Wiedner Hauptstraße 8, 1040 Vienna, Austria

**Correspondence:** Wolfgang Preimesberger (wolfgang.preimesberger@geo.tuwien.ac.at)

**Abstract.** The ESA CCI Soil Moisture multi-satellite climate data record is a widely used dataset for large-scale hydrological and climatological applications and studies. However, data gaps in the record can affect derived statistics such as long-term trends, and – if not taken into account – can potentially lead to inaccurate conclusions. Here, we present a novel gap-free dataset, covering the period from January 1991 to December 2023. Our dataset distinguishes itself from other gap-filled products, as it is purely based on the available soil moisture measurements (independent of ancillary variables to make predictions), and further due to the inclusion of uncertainty estimates for all interpolated data points.

Our gap-filling framework is based on a well-established univariate Discrete Cosine Transform with Penalized Least Squares (DCT-PLS) algorithm. This ensures that the dataset remains fully independent of other soil moisture and biogeophysical datasets, and eliminates the risk of introducing non-soil moisture features from other variables. We apply DCT-PLS on a spatial moving window basis to predict missing data points based on temporal and regional neighbourhood information. The challenge of providing gap-free estimates during extended periods of frozen soils is addressed by applying a linear interpolation for these periods, which approximates the retention of frozen water in the soil. To quantify the inherent uncertainties in our predictions, we developed an uncertainty estimation model that considers the input observations quality and the performance of the gap-filling algorithm under different surface conditions. We evaluate our algorithm through performance metrics with independent in situ reference measurements and by its ability to restore GLDAS Noah reanalysis data in artificially introduced satellite-like gaps. We find that the gap-filled data performs comparable to the original observations in terms of correlation and ubRMSD with in situ data (global median $R = 0.72$, $ubRMSD = 0.05\ m^3m^{-3}$. However, in some complex environments with sparse observation coverage, performance is lower.

The new ESA CCI SM v09.1 *GAPFILLED* dataset is publicly available at https://doi.org/10.48436/s5j4q-rpd32 (Preimesberger et al., 2024) and will see yearly updates due to its inclusion in the operational ESA CCI SM production.

## 1 Introduction

Satellite soil moisture data are relevant for a wide range of applications, including water resource management, agriculture, disaster risk assessment and response, weather prediction, and climate monitoring (Dorigo et al., 2021a; Srivastava, 2017). Soil moisture (SM) is classified as an essential climate variable (ECV) (GCOS, 2022) and dedicated satellite missions such as ESA's SMOS (Kerr et al., 2010) or NASA's SMAP (Entekhabi et al., 2010) have been launched to measure it. However, the



data coverage of individual satellites is limited by their lifespan and revisit frequency, requiring multiple days for a complete set of global measurements. Long-term analyses of climate variability and change require at least 30 years of data (GCOS, 2022), which surpasses the lifespan of any single Earth observation satellite. Measurements from multiple platforms are therefore harmonized and merged to create consistent multi-decadal records, such as the ESA CCI soil moisture (ESA CCI SM) products

(Dorigo et al., 2017; Gruber et al., 2019; Preimesberger et al., 2021). Despite the high number of currently operational satellites integrated in ESA CCI SM, it is still impossible to provide global gap-free daily soil moisture, because of several physical limitations to measure soil moisture, as discussed below. Leaving these data gaps unfilled can perturb derived statistics such as anomalies or long-term trends, and in the worst case lead to incorrect conclusions (Bessenbacher et al., 2023; Liu et al., 2020c). Yet, observational datasets like ESA CCI SM are still valued in addition to gap-free reanalysis products because they

can capture features that models might miss (van der Schalie et al., 2022). This creates a challenge, as gap-free satellite data are often required, especially in various machine learning frameworks, such as used for spatial downscaling (Kovačević et al., 2020; Nadeem et al., 2023), or for applications that require water balance closure (Dorigo et al., 2021a). In an attempt to fill these gaps, users unfamiliar with the underlying geophysical processes may resort to oversimplified approaches, such as filling missing values ("NaNs") with zeros, which risks introducing further bias into the analysis. Understanding the underlying

causes of data gaps in ESA CCI SM is therefore a prerequisite for filling them.

As mentioned above, there are several reasons for data gaps in ESA CCI SM. Apart from missing satellite overpasses in earlier periods, data gaps originate from the underlying retrieval models to convert raw satellite measurements (radiances, backscatter) into soil moisture. These models become unreliable under certain surface conditions, leading to flagged/masked data points and, therefore, gaps in the final record. The most common cause are frozen soils (van der Vliet et al., 2020).

Retrieval models are designed to estimate the amount of liquid water in the soil (Owe et al., 2008; Wagner et al., 1999; Entekhabi et al., 2010). The dielectric properties of water change drastically between liquid and frozen aggregate state (Naeimi et al., 2012). Consequently, measured soil moisture normally drops when part of the satellite footprint is frozen (Amankwah et al., 2021; Dorigo et al., 2021b). The second most common cause for unreliable retrievals is dense vegetation, which can mask contributions from the soil. This issue is more pronounced at shorter wavelengths (K- and X-band), while L and C-band

measurements are less affected (van der Schalie et al., 2021; Jackson et al., 1982). Other reasons for failed retrievals include signal perturbation due to radio frequency interference (RFI), mainly for C and L-band and over densely populated areas (Uranga et al., 2022; Oliva et al., 2012), as well as barren, dry soils where the received signal is noise-like and sometimes dominated by responses from sub-surface sediments (Wagner et al., 2024; Petropoulos et al., 2015).

Data gaps can be broadly classified into two – not strictly separable – categories (based on Bessenbacher et al. (2022); Rubin

(1976); Dorigo et al. (2017)): (i) (quasi-)random and (ii) systematic retrieval gaps. Random gaps appear noise-like over time and affect only short periods. They are not directly related to processes at the surface. Members of this category include cases where there is no satellite overpass for a given day, when the retrieval model does not converge for various reasons, or when it converges at soil moisture values outside the physically possible boundaries, and RFI. Systematic gaps are related to the surface state and, therefore, usually cover longer periods. In the most extreme cases, systematic gaps can be permanent, meaning no

data is available for a location (e.g., rainforests in ESA CCI SM). Systematic gaps are more likely to obscure spatial and/or





temporal features in the data than random gaps. Gap-filling methods aim to restore these features as accurately as possible, without introducing artificial (non-soil moisture) patterns and without overfitting the data.

There are generally two approaches to fill gaps in satellite soil moisture records: (i) univariate, stand-alone interpolation methods, and (ii) multivariate, covariate-enhanced interpolation methods. The former uses only the available (neighbourhood) information to predict missing values with (geo)statistical methods, while the latter relies on supporting data to improve predictions. For soil moisture, most recent gap-filling studies focus on the applicability of different multivariate machine learning (ML) methods in small to medium-sized study regions. Random forest (RF) has been the dominant algorithm (Nadeem et al., 2023; Liu et al., 2020c; Nadeem et al., 2023; Wang et al., 2023; Mao et al., 2019; Bessenbacher et al., 2022), which was found to outperform other covariate supported approaches such as Neural Networks (NN), XGBoost, Support Vector Machines (SVM), or Multivariate Linear Regression (Liu et al., 2023; Sun and Xu, 2021; Tong et al., 2021; Almendra-Martín et al., 2021). To predict missing soil moisture, these studies use physically related variables such as air or land surface temperature, precipitation, soil type, topography, land cover, and vegetation properties. While multivariate approaches can leverage additional datasets to improve predictions, univariate methods such as ordinary kriging (OK) or multiple linear regression (MLR) do not. However, they remain viable alternatives. In homogeneous areas, they have often been found to perform similarly to multivariate methods (Tong et al., 2021; Sun and Xu, 2021; Yang et al., 2018).

In this study, we adopt a univariate approach to fill gaps in ESA CCI SM. The choice of using a univariate method is motivated by the fact that it allows for being totally independent of any ancillary data, and therefore mitigating the risk of introducing spurious features. The greatest possible independence of ancillary or model data is also one of the top priorities expressed by the climate user community (Dorigo et al., 2017). Gap-filling of ESA CCI SM is based on Discrete Cosine Transform with Penalized Least Squares (DCT-PLS) (Garcia, 2010), because (i) even without the use of any ancillary data, it often performs similarly to multivariate methods such as XGBoost, LSTM, ANN or CNN (Shangguan et al., 2023; Yang et al., 2018). (ii) It is considered a well-proven method that has been successfully applied to gap-fill soil moisture (Wang et al., 2012; Yang et al., 2018) and other geophyiscal variables such as LST (Liu et al., 2020a; Pham et al., 2019), ocean surface current (Kongkulsiri et al., 2018), chlorophyll-a concentration (Wang et al., 2022), and lake surface water temperature (Fan et al., 2022) data.

The multitude of gap-filling studies highlights the importance of this topic and the community's need for a gap-free ESA CCI SM data record. With the ESA CCI SM v09.1 *GAPFILLED* product, we aim to meet this community requirement by providing an independent, gap-filled, global, long-term, satellite-based record. Consistent with other ESA CCI SM products, we also provide uncertainty estimates to quantify potential errors associated with the interpolation process — a feature which to our knowledge has not yet been included in any gap-filled satellite soil moisture record. This will allow users to account for the expected accuracy of our predictions and, for example, weigh them accordingly in their analyses.





## 2 Data

### 2.1 ESA CCI SM

ESA CCI SM is a multi-satellite climate data record of daily global surface soil moisture. The latest release, v09.1, covers
the period from 1 November 1978 to 31 December 2023 (Dorigo et al., 2017; Gruber et al., 2019; Preimesberger et al., 2021;
Dorigo et al., 2024a). The COMBINED product merges soil moisture derived from radiometer measurements in the L-, C-, X-,
and K-bands by the Land Parameter Retrieval Model (LPRM) (Owe et al., 2008; van der Schalie et al., 2015; der Schalie et al.,
2016; van der Schalie et al., 2021) and from C-band scatterometer observations using the TU Wien change detection method (H
SAF, 2022; Wagner et al., 1999, 2013; Hahn et al., 2021). A timeline of all used satellite sensors and frequency bands is shown
in Fig. 1a. Soil moisture values are provided in volumetric units ($m^3 m^{-3}$). ESA CCI SM data include quality flags, which are
applied to mask unreliable data points and inform users about the underlying cause. Currently, there are quality flags for (i)
frozen soils, (ii) dense vegetation, (iii) unsuccessful retrievals in all sensors, (iv) low Signal-to-Noise in all sensors, and (v)
barren ground (van der Vliet et al., 2020; Parinussa et al., 2011). While flags (i)-(iv) coincide with missing data points in ESA
CCI SM, flag (v) only tags the observations but does not remove them. ESA CCI SM data come with associated estimates of
measurement uncertainties (Gruber et al., 2017, 2019). These uncertainties are propagated from sensor-level estimates through
triple collocation analysis on a day-of-year basis (Stradiotti et al., 2024). ESA CCI SM represents moisture in the upper soil
layer (~0-5 cm) with a regular spatial sampling of 0.25 degrees (~25 km). The algorithm is regularly updated with new data
from available sensors to incorporate the latest scientific advances and extend the product's temporal coverage. This study
aims to fill data gaps in the period from 1 January 1991 onward, chosen to make the data suitable for long-term (multi-decadal)
studies (GCOS, 2022). From 1978 to 1991, no overlapping sensors were available (Fig. 1a), resulting in very low data coverage
(Fig. 1b) and high measurement uncertainties (Dorigo et al., 2017). As this would lead to even higher uncertainties after gap-
filling, this period is excluded.





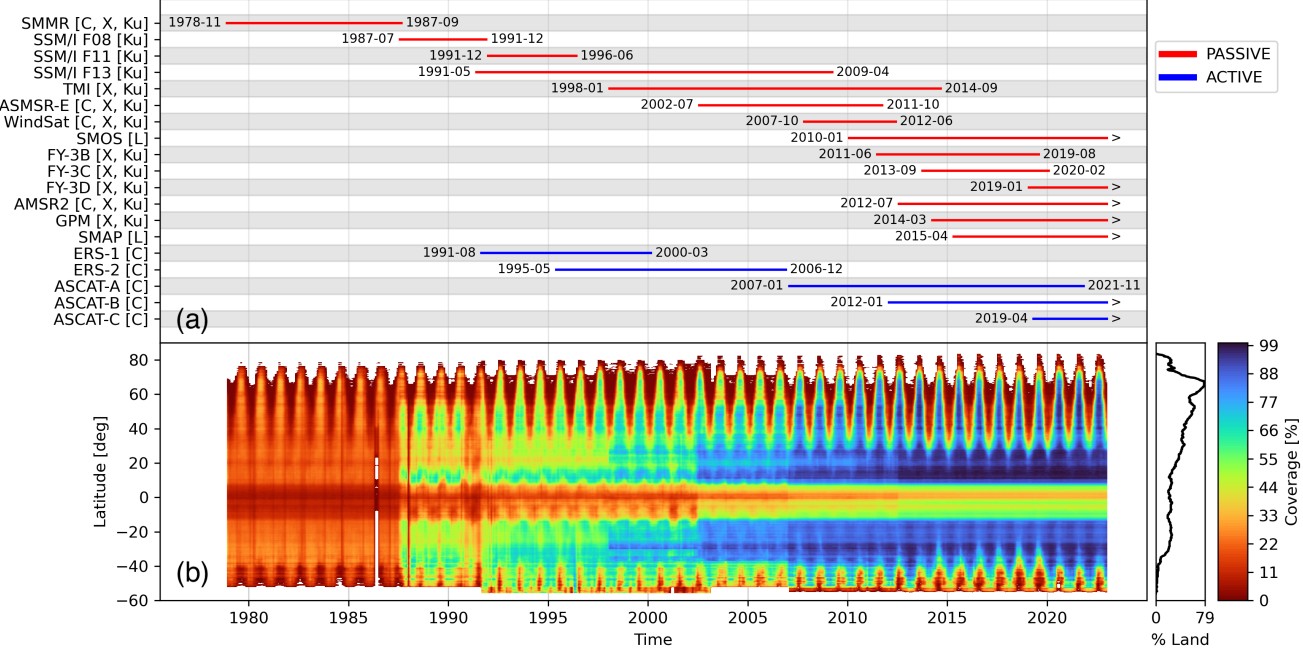

**Figure 1.** Availability of satellites and sensor frequency bands over time as of ESA CCI SM v09.1 (a), and monthly data coverage over time of the merged COMBINED product averaged by latitude (b).

## 2.2 Model and reanalysis data

We use daily mean top-layer (0-7 cm) soil temperature from ERA5 at a 0.25 degree resolution (Hersbach et al., 2020). In our
study, these data are used to classify which gaps occurred during periods of frozen soils and which not, and to subsequently choose one of two different interpolation approaches. ERA5 is a global reanalysis product that integrates in situ and satellite observations into an open-loop run of the HTESSEL land surface model, producing hourly, gap-free estimates of various surface variables since 1940.

We further use ERA5-Land top-layer (0-7 cm) soil moisture from 1991 to 2023 for a comparison of zonal anomalies and
long-term trends after gap-filling (Muñoz Sabater et al., 2021). ERA5-Land has been produced by replaying the land component of ERA5, and provides surface variables with an increased spatial resolution (0.1 degree).

Additionally, as reference data to assess the impact of gap distribution and size on interpolation quality, we use daily averages of gap-free soil moisture simulations for the 0-10 cm layer (given in $\mathrm{kg\,m^{-2}}$) from the GLDAS Noah v2.1 model (Rodell et al., 2004). The original data are available at a 3-hourly temporal sampling and 0.25 degree spatial resolution for the period after
2000. Notably, GLDAS Noah surface soil moisture serves as the scaling reference for the ESA CCI SM COMBINED product (Dorigo et al., 2017). Hence, there is no bias between the two.





## 2.3 Vegetation optical depth

We utilize Vegetation Optical Depth (VOD) from the VODCA v2 CXKu dataset (Zotta et al., 2024; Moesinger et al., 2020) to predict the uncertainty of gap-filled values under different vegetation conditions. VOD is a satellite-derived, unitless measure of vegetation density, closely related to vegetation water content and biomass. VODCA v2 CXKu harmonizes and merges VOD retrievals from 9 passive sensors operating in the 6.8-19.4 GHz frequency range, creating a consistent long-term record from 1987 to 2023.

## 2.4 In situ measurements

The International Soil Moisture Network (ISMN) is a data hosting facility that collaborates with partners worldwide to collect in situ measurements of soil moisture and other geophysical variables into a harmonized, quality-controlled database (Dorigo et al., 2021b, 2013, 2011). ISMN in situ measurements serve as the primary source of ground reference data for global satellite soil moisture validation activities. An overview is provided by Dorigo et al. (2021b). In our study, we use ISMN data to evaluate the quality of our gap-filled soil moisture product. We use a subset of the full ISMN database for the 0-10 cm depth range, downloaded in March 2024. Specifically, we use a subset of the available data labelled as Fiducial Reference Measurements (FRMs), which are point measurements found to be representative of soil moisture at the radiometer scale, and therefore recommended for satellite validation (Himmelbauer et al., 2023; Goryl et al., 2023). This subset consists of 1,314 high-quality time series (Appendix Fig. A2). Soil moisture measurements are provided with associated quality flags for each time stamp and station metadata, such as land cover information extracted from ESA's CCI land cover v1.6.1 dataset. ISMN measurements are unevenly distributed both temporally (with different stations covering different periods) and geographically. This uneven distribution means that our validation results are spatially and temporally biased towards regions and periods with dense station coverage, primarily North America and Europe after the year 2000 (Dorigo et al., 2021b).

## 3 Preprocessing

The following steps are applied to the original ESA CCI SM v09.1 COMBINED data before the interpolation process takes place.

## 3.1 Additional retrievals under dense vegetation conditions

Soil moisture values in areas covered by tropical rainforests are associated with high uncertainties (Ulaby et al., 2014). Consequently, these areas are masked out in ESA CCI SM, leading to permanent gaps (Dorigo et al., 2017). Such large gaps are particularly difficult to fill for any algorithm, regardless of whether covariates are used or not (Tong et al., 2021). However, soil moisture is still retrieved, e.g. from deforested patches, and therefore available for some frequency bands in which electromagnetic waves can - at least to some degree - penetrate through vegetation. Specifically, C-band data from ASCAT-A, B, and C, as well as L-band data from SMOS and SMAP - despite the overall lower data quality compared to other global regions - can





serve as useful support points for the interpolation algorithm when available. Hence, these lower-quality observations, despite being masked out in the original ESA CCI v9.1 COMBINED data, are included now in the gap-filling to serve as anchor points (compare Fig. 2e). This is preferred over large-scale interpolation without any support data (Guo et al., 2022).

Similar to the rest of the ESA CCI SM data, the additional retrievals available for densely vegetated areas are error-characterized via triple collocation analysis, harmonized through scaling to a common reference, and merged when more than one measurement is available for a grid point on a given day (Liu et al., 2012; Dorigo et al., 2017; Gruber et al., 2017, 2019).

### 3.2   Gap analysis

The observation density of ESA CCI SM depends on the number of available satellites and surface conditions that affect
retrieval success rates. Fig. 1b shows the temporal coverage of the product. At the start of the data record, data from only few sensors were available, leading to relatively low data coverage. After 2002, coverage increases significantly, culminating in up-to-daily (i.e., gap-free) measurements for certain latitude bands in recent years. However, some regions still show large data gaps (Fig. 2e), especially around the equator and above 50° N, due to dense vegetation cover and seasonally frozen soils, respectively.

Figures 2a-d evaluate data gaps in ESA CCI SM in terms of their three-dimensional (Euclidean) distance from the nearest available valid data point. Gaps are analysed separately depending on whether the corresponding reanalysis soil temperature is above or below 0 °C. While in most cases a valid observation is found within a range of a few pixels or days, in some exceptional cases, the combined gap size in days and pixels can reach up to 350.

The so-derived distances are required for the initial guess for gap-filling (nearest neighbour) which is then further adjusted
by the DCT-PLS. They are also essential for deriving uncertainty estimates.



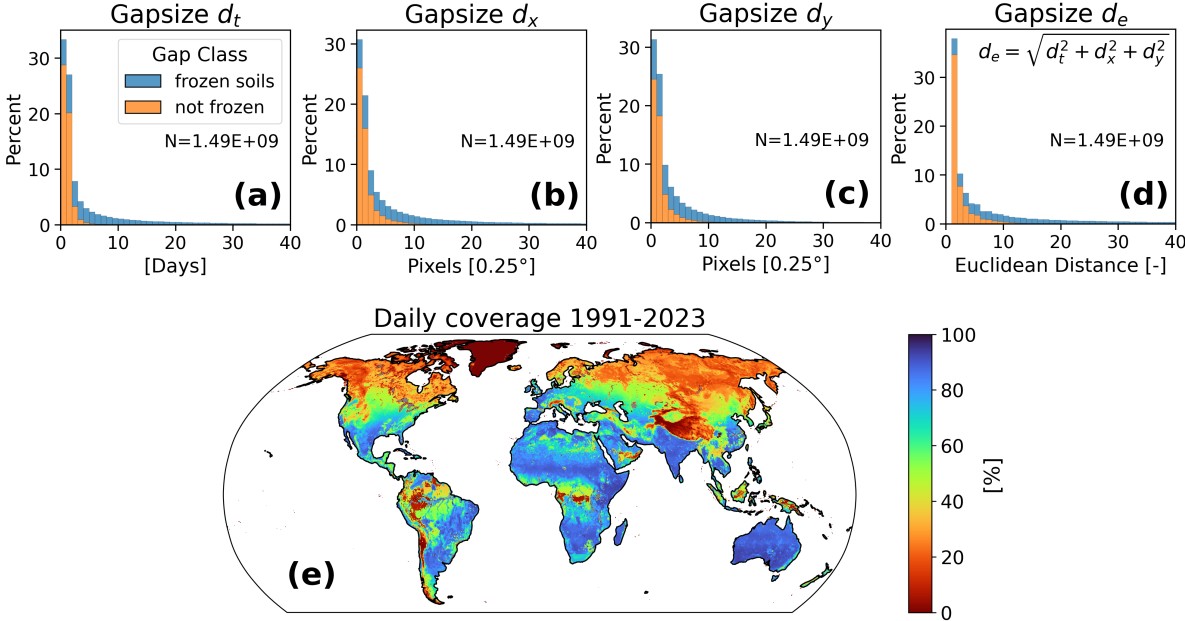

**Figure 2.** Distance to the nearest neighbour for each missing data point (excl. Greenland) in ESA CCI SM along each dimension: time (a), longitude (b), latitude (c), and Euclidean distance across dimensions (d). Very large gaps (40-350 days/pixels) exist, but are too rare to be visible in the histograms. The temporal coverage for each grid point with additional anchor observations over tropical rainforests is shown in (e). Gap classification (frozen vs. non-frozen) based on ERA5 soil temperature.

## 4   Gapfilling Methods

To fill data gaps between observations in ESA CCI SM and assign uncertainty estimates to the predicted values, we follow the processing chain in Fig. 3. We start from daily ESA CCI SM data with additional observations in the tropics (Sect. 3.1). We then compute the gap statistics (Sect. 3.2) required for all subsequent steps: (i) for each gap we compute the distance to the

nearest observation (across 3 dimensions), (ii) we differentiate between gaps due to frozen soils and other reasons, based on (gap-free) ERA5 temperature information. We then fill data gaps in ESA CCI SM by applying the DCT-PLS algorithm with local parameterization (Sect. 4.1) but overwrite the so-derived values with a linear interpolation over time in periods where soil moisture is frozen in a certain place (discussed in Sect. 4.1.1). Prediction uncertainty models are derived based on the performance of our method in restoring GLDAS Noah soil moisture data after introducing gaps from ESA CCI SM (Sect. 5).

Finally, the gap-filled soil moisture and uncertainty estimates from gap statistics and pre-computed models are combined in the final *GAPFILLED* product.





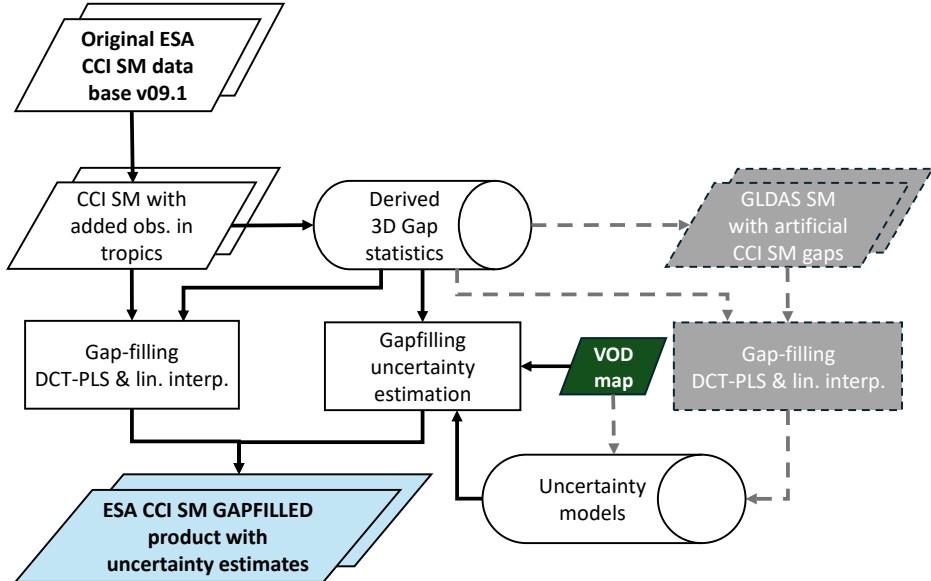

**Figure 3.** Flow chart of the main process steps to derive the ESA CCI SM *GAPFILLED* product (blue box). Dashed lines indicate optional processing steps, that - once the model parameters are known - can be left out.

## 4.1 Core algorithm: DCT-PLS

We only provide a summary of the DCT-PLS algorithm here. For the full description with all intermediate steps, see Garcia (2010). Initially designed as a data smoother, DCT-PLS can also provide estimates for missing data points by predicting a set of smoothed values $\hat{y}$ for the original / input data $y$. The algorithm optimizes for (i) minimal Residual Sum of Square (RSS) between the input and smoothed observations and (ii) optimal reduced roughness $P(\hat{y})$ between (smoothed) elements (Eq. 1). DCT-PLS therefore ideally removes noise in the data while retaining relevant information, i.e. finding smoothed versions of input observations $y$ so that $F(\hat{y})$ is minimised.

$$F(\hat{y}) = RSS + sP(\hat{y}) = ||y - \hat{y}|| + sP(\hat{y}) \tag{1}$$

The penalty term $P(\hat{y})$ is based on (temporal and spatial) neighbourhood information as differences ($D$) between elements of $\hat{y}$, thus optimizing for smoothed transitions between them (Eq. 2).

$$P(\hat{y}) = ||D\hat{y}||^2 \tag{2}$$

Solving the linear system (Eq. 3, where $I_n$ indicates the Identity matrix) to minimize $F(\hat{y})$ is a computationally extensive task when $y$ is large.





$$(I_n + sD^T D)\hat{y} = y \tag{3}$$

Garcia (2010) provides a step-by-step description of the required (matrix) operations to find $\hat{y}$ and discusses modifications to the base algorithm for improved performance. In particular, computing $D$ and performing Eigenvalue ($\lambda$) decomposition thereof is greatly simplified by the use of equally spaced input data such as ESA CCI SM. In fact, a predefined formulation for $\lambda_i$ of $D$ (Eq. 4) can be used in the 3-dimensional case ($N = 3$) to build the tensor $\Lambda$ in Eq. 5. Together with a realization of the smoothing parameter $s$, we can define the tensor $\Gamma^N$ (Eq. 6).

$$\lambda_i = -2 + 2\cos(i-1)\pi/n \tag{4}$$

$$\Lambda^N_{i_1,\dots i_N} = \sum_{j=1}^{N} \left( -2 + 2\cos\frac{(i_j-1)\pi}{n_j} \right) \tag{5}$$

$$\Gamma^N = 1^N \div (1^N + s\Lambda^N \circ \Lambda^N) \tag{6}$$

This allows to efficiently solve for $\hat{y}$ using the discrete cosine transform matrix (DCTN) of $y$, and the inverse form (IDCTN) for $N = 3$ dimensional data respectively (Strang, 1999). As data gaps are present in ESA CCI SM, assigning weights ($W$) to observations is required (Eq. 7). Data gaps are assigned a weight of zero, and therefore interpolated as part of the (robust) smoothing process, while observations have an initial weight of 1.

$$\hat{y}_{k+1} = IDCTN\left(\Gamma^N \circ DCTN(W \circ (y - \hat{y}_k) + \hat{y}_k)\right) \tag{7}$$

Finally, having an estimate for $\hat{y}$, the generalized cross-validation (GCV) score is computed (Eq. 8), where $n_{miss}$ is the number of missing values of $n$ overall samples.

$$GCV(s) = \frac{wRSS/(n-n_{miss})}{(1-Tr(H)/n)^2} = \frac{||W^{1/2}(\hat{y}-y)||^2/(n-n_{miss})}{(1-Tr(H)/n)^2} \tag{8}$$

$Tr(H)$ is computed from $\lambda$ (Eq. 4), as described in Eq. 9.

$$Tr(H) = \sum_{i=1}^{n} [1 + s(2 - 2\cos((i-1)\pi/n))^2]^{-1} \tag{9}$$

A bounded minimization – as implemented in SciPy's "optimize" module (Virtanen et al., 2020) – is now applied to find the smoothing parameter $s$ for minimal $GCV$, and $\hat{y}$ is computed again using this $s$. $s$ is the only free parameter in the model that needs to be tuned using GCV.



As outliers can be present in ESA CCI SM data, we use the "robust" implementation of DCT-PLS (Garcia, 2010), which includes further iteration of the above described process to detect outliers in the data and gradually reduce weights assigned to these observations until the optimization converges at $\hat{y}_{k+1} - \hat{y}_k \to 0$, meaning that no more relevant changes between $y$ and $\hat{y}$

- before and after calibrating $s$ respectively - are found.

DCT-PLS can take n-dimensional tensors of any size. We apply it to (3-dimensional) spatial subsets of the dataset via a $15 \times 15$ degree moving window. We prefer a moving window over a global $s$-value to locally allow different degrees of smoothing and therefore reduce potential overfitting or over-smoothing when using a global $s$-value (Wang et al., 2012). Therefore, local parameterization of $s$ is also expected to improve the interpolation results between areas with different levels of

autocorrelation. In most cases, the chosen window is large enough to provide sufficient data for a robust estimation. However, for some edge cases, such as remote islands, the initial window size is gradually extended by 5 degrees (in 8 directions, in practice the maximum was 35 degrees), until the GCV minimization process has sufficient data to converge for $s$.

Predictions are not only made for missing data points, but also at locations where observations are available. While these (smoothed) observations are not used in the final gap-filled product (because the original observations are kept when available),

they can be used to further harmonize predictions and observations before they are combined (see Sect. 4.1.2).

### 4.1.1 Frozen soils

During periods of frozen soil moisture, it is a matter of definition, whether a dataset should represent only the liquid, or the total (frozen and non-frozen) soil moisture content. In the first case, soil moisture would start to decline as soon as water begins to freeze in a scene, until reaching ~0 $m^3 m^{-3}$. Factors such as freezing point depression due to solutes can result in the

presence of liquid soil moisture even when the soil temperature is below 0 °C (Amankwah et al., 2021). In the second case, which we try to cover in our data record, soil moisture remains constant over the period when it is frozen, i.e. there is no soil water loss due to evaporation of percolation. However, when predicted observations are based on insufficiently-masked, low and often noisy soil moisture observations from transitional zones/periods around a frozen soil gap, these are often too low (Wang et al., 2012; Liu et al., 2023). In addition, since DCT-PLS does not use any information on soil temperature, it may

predict temporal fluctuations in soil moisture even when the soil is frozen. One could argue that this is the result of regional (sub-pixel) freeze/thaw processes, but there is no reason to assume that univariate algorithms, which do not account for soil temperature on a sub-pixel scale, can accurately predict this.

We therefore carry the last (observed or interpolated) soil moisture level before freezing starts forward over time until thawing occurs and soil moisture changes can be measured again directly by the satellite. In practice, there can be differences

between the last and first available observations around a period of frozen soil, e.g., due to measurement noise. We therefore use the mean of 30 days before and after an affected period to perform a simple linear interpolation over time between these points. This is in line with our definition of frozen soil moisture content, deals with the expected noise in observations around the affected period, and avoids temporal fluctuations in predictions when soil moisture should remain stable. These values are then used to replace those from the DCT-PLS algorithm.





### 4.1.2 Combining predictions and observations

For the final *GAPFILLED* product, we use the predictions (from either DCT-PLS or linear interpolation) to fill data gaps in the original record. The available original observations are not replaced. However, since DCT-PLS uses temporal and spatial neighbourhood information to create predictions, the mean and variance of the predictions for a grid point are adjusted towards those of its neighbours, giving the data a (spatially) smooth appearance. This characteristic, typical of any spatial interpolation method (Llamas et al., 2020), can lead to temporal inconsistencies between predictions and observations. To address this, we apply a linear transformation (derived from linear regression on shared data points) to scale the mean and variance of the predictions to match those of the observations (Steven et al., 2003; Paulik et al., 2024). Finally, the scaled predictions are used to fill gaps in the observation time series.

### 4.2 Validation

The gap-filled ESA CCI SM product is compared to ISMN in situ reference measurements using the QA4SM online validation platform (https://qa4sm.eu). Time series for grid points collocated with FRMs (see Appendix Fig. A2) are extracted from the original ESA CCI SM COMBINED and the *GAPFILLED* product. For the latter we extracted four different subsets: (i) All available data points from the *GAPFILLED* product (original observations plus filled values), (ii) only the filled values, (iii) only the filled values from "small" gaps (classification based on the Euclidean distance (ED) to the nearest observation) with $ED < 2$ (i.e., adjacent to a valid data point in any dimension), (iv) only the filled values within "medium/large" gaps ($ED \geq 2$). All four datasets were uploaded to the service. We select a temporal matching window of $\pm 1$ hour around 00:00 UTC to align in situ measurements with the satellite data (temporal reference). QA4SM offers filtering options to include only in situ time stamps flagged as "good" – excluding erroneous measurements from e.g. frozen soils (Dorigo et al., 2013) – and "(very) representative" time series in terms of FRM qualification. Biases between satellite and in situ time series are removed by matching their mean and standard deviation, using the in situ data as the (scaling) reference (Gruber et al., 2020). QA4SM validation results include ESA CCI land cover information at the in situ sites, which we use to stratify our results.

## 5 Uncertainty estimation

An uncertainty estimate is provided for each interpolated value. This uncertainty depends on the uncertainty of the observations used ($\sigma_{observations}$) and the availability of support data and other inherent factors affecting the quality of DCT-PLS predictions (summarized as $\sigma_{gapfilling}$). Combining both as in Eq. 10 yields the uncertainty for our predictions.

$$\sigma_{prediction} = \sqrt{\sigma_{observations}^2 + \sigma_{gapfilling}^2} \tag{10}$$

$\sigma_{observations}$ is characterized in ESA CCI SM using triple collocation analysis and uncertainty propagation (Gruber et al., 2016). The random error level of observations mainly depends on the merged sensor frequency bands in conjunction with surface characteristics such as vegetation cover (Parinussa et al., 2011). Recently, a temporal component was added to account



for (sub)seasonal error dynamics (Stradiotti et al., 2024). We use $\sigma_{observations}$ as the baseline for uncertainties of the gap-filled values. For a conservative estimate, we choose the 95th percentile of measurement uncertainties for any grid point series (or the nearest neighbour series in the case of a permanent gap).

To estimate $\sigma_{gapfilling}$, we impose the gaps in the satellite data on gap-free GLDAS-Noah soil moisture and subsequently restore these gaps using the presented methodology (Fig. 3). This approach preserves the original satellite gap systematics,
as opposed to randomly splitting the input data into training and validation sets, as is commonly done (e.g., Wang et al. (2012); Liu et al. (2020b); Zhang et al. (2021)). The goal of this analysis is to derive a set of model functions for the expected discrepancy between predictions and observations based on the gap size. Uncertainty models are preferred over empirical values because they remain applicable even in scenarios where GLDAS data are unavailable, ensuring broader utility of the methodology (e.g., for the period before 2000, when GLDAS-Noah v2.1 data are not available). For improved accuracy, these
functions should account for both realistic gap conditions and varying surface complexities. To address the former, we use the previously computed gap statistics (Euclidean distances). To address the latter, we use a VOD classification map for low, medium, and high VOD derived from the VODCA v2 CXKu archive (compare Appendix Fig. A1). We therefore assume that prediction uncertainty will generally increase for larger gaps and with denser vegetation cover, consistent with the observations (Parinussa et al., 2011).

From analysing the differences between original and restored values with regard to the gap systematics and VOD level (compare Fig. 4a), a log-like function of form $f(x) = b \cdot (1 - e^{a \cdot x})$ is fitted for different vegetation conditions. Each $a$ and $b$ is found from fitting 1000 realizations of this function (least squares with boundary conditions for $a < 0$ and $b > 0$) to randomly drawn samples without replacement. Using the medians of the obtained parameter sets for $a$ and $b$ (Table 1), we defined the required models for the uncertainty of the gap-filling process ($\sigma_{gapfilling}$) itself, as shown in Fig. 4b.

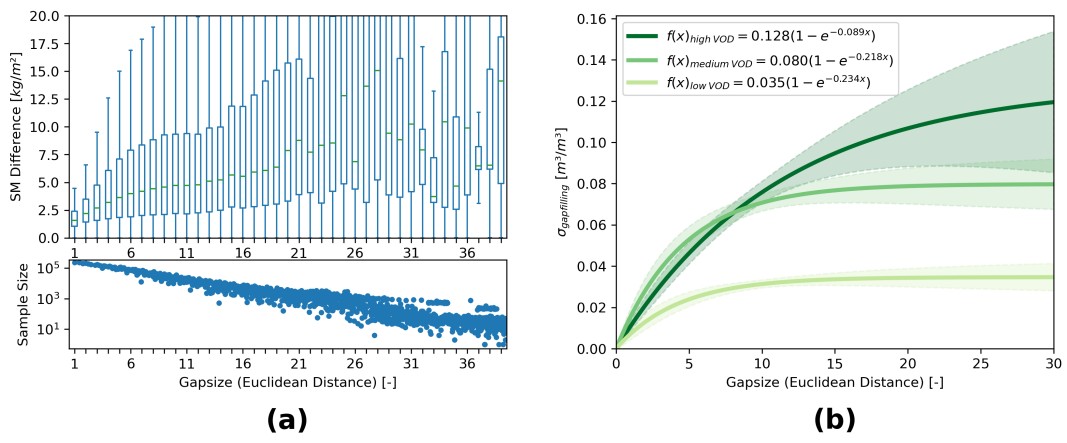

**Figure 4.** Absolute differences between original and restored GLDAS Noah soil moisture after gap filling (a), and function fits (b) (median and standard deviation, from 1000 randomly drawn samples) to predict $\sigma_{gapfilling}$ for high, medium, and low VOD conditions.





**Table 1.** Model parameters found from fitting f(x) for different VOD levels (VOD classification in Appendix Fig. A1).

| $f(x) = b \cdot (1 - e^{a \cdot x})$ | a | b |
|---|---|---|
| high VOD | -0.089 | 0.128 |
| medium VOD | -0.218 | 0.08 |
| low VOD | -0.234 | 0.035 |

## 6  Results

### 6.1  DCT-PLS parameterization

Fig. 5 shows the result of the local parameterization of the DCT-PLS model. A lower $s$ means that more (day-to-day temporal and/or regional spatial) variability is found in the prediction (Fig. 5a). The highest $s$ values are found in arid climates, in NE Africa and Arabia. This coincides with the expected low variability of soil moisture in these regions. Low $s$ values (i.e. high variability) are found in subtropical regions in Central Africa and South America, South Asia and East Australia. The GCV-Scores in Fig. 5b were minimized to find the optimal $s$ in each cell. For most regions, the GCV-Score is close to zero, which indicates a good fit. Higher scores are found in sub-tropical regions, which indicates a slightly larger discrepancy between the predictions and available observations in these areas. This could be due to underestimating the seasonal and/or overfitting of the day-to-day soil moisture variability in the measurements.

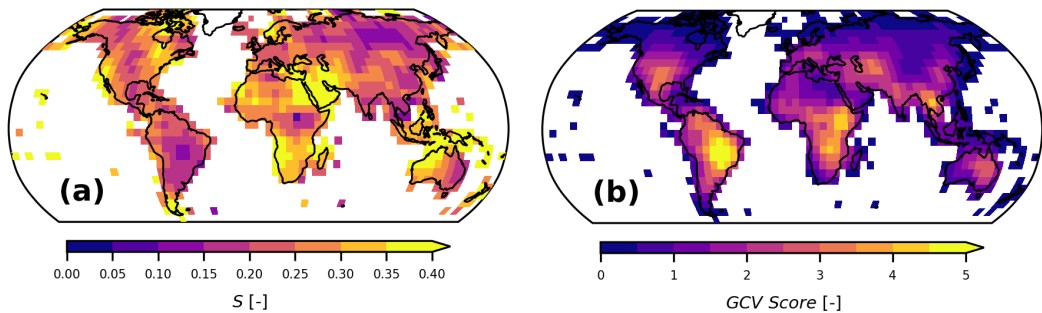

**Figure 5.** Local parameterization of the DCT-PLS function to find the optimal value for $s$ (a) that minimizes the GCV-score (b).

### 6.2  Spatial data characteristics

The *GAPFILLED* product provides full daily coverage for both soil moisture and uncertainty estimates. For a selected day, Fig. 6 shows both fields, before and after gap-filling respectively. The gap-filling has produced a smooth, spatially consistent image, without any outlier values or edges that could arise from sudden differences in regional parametrization of the DCT-PLS model. As expected, soil moisture in tropical latitudes is high, as a result of interpolating anchor retrievals from L- and C-band

measurements within and surrounding these areas. In high northern latitudes, the predictions come mainly from linear time interpolation between autumn values of the preceding year and spring values observed later in the year. The uncertainties are within the range of 0 to 0.1 $m^3m^{-3}$, which is expected from the defined models for $\sigma_{gapfilling}$. The highest values are found in (sub)tropical and boreal ecozones, the Tibetan plateau, and SE-Asia. This is due to the low data coverage and quality of the used retrievals in these regions, which are noisy for dense vegetation and often missing in mountainous regions.

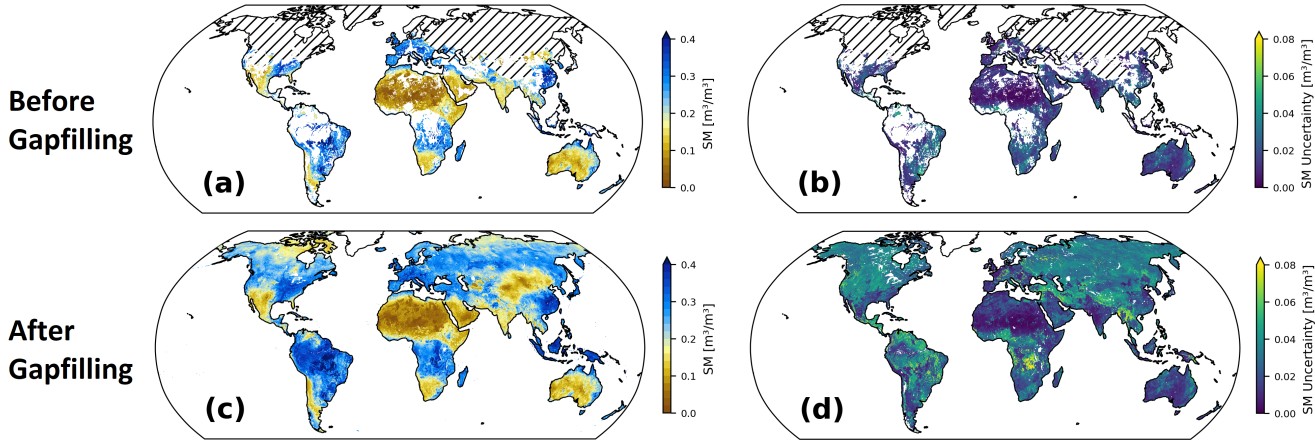

**Figure 6.** Soil moisture and uncertainty before (a), (b) and after (c), (d) gap-filling on 1 March 2007. Ice sheets (Greenland, Antarctica) are excluded from the *GAPFILLED* product. Hatched regions indicate ERA5 soil temperature below 0 °C.

### 6.2.1   Time series characteristics

Fig. 7 shows temporal subsets for five grid points in the Northern Hemisphere, where also in situ measurements are available. The locations were selected to represent a wide range of environments and gap types, as far as the spatial distribution of in situ sensors allows (for more information on the chosen sites, see Appendix Table A1). For visualization purposes, a linear least-squares regression scaling was applied, to remove biases between point- and satellite-scale time series (Steven et al., 2003; Paulik et al., 2024). Anomalies are computed relative to the 1991-2023 average conditions. All R-scores are based only

on data points from the visualised period and are statistically significant ($p < 0.05$). Fig. 7a illustrates the performance of the gap-filling algorithm in predicting soil moisture for gaps because of frozen conditions and gaps emerging for other reasons in an area with a moderate number of missing data points (40 %). The predicted values align reasonably well with the available in situ data, with an anomaly correlation of 0.59 between the in situ measurements and the filled values only ($R_{FILLEDVALUES}$,

without the original observations). This is lower than the 0.70 correlation between in situ measurements and original satellite observation anomalies. However, this is hardly surprising, as systematic gaps appear during more challenging conditions. Fig. 7b is an example for a location affected strongly by seasonally missing data. The periods of linear interpolation (frozen) align well with missing in situ values due to the (independent) quality-flag-based masking based on in situ temperature measurements (Dorigo et al., 2021b). The reanalysis temperature data below 0 °C therefore for this location provides a good approximation





of the period when soil moisture is frozen and kept at an almost constant level. However, it also means that a direct comparison between gap-filled values and in situ measurements is usually not possible during Winter. For the onset and termination of this period, the absolute values between gap-filled and in situ data match well. Fig. 7c represents a location with only few missing data points and no relevant periods of frozen soil moisture. In this case, we find a higher correlation with the in situ data for the gap-filled values compared to the original data (0.89 vs. 0.80), which we attribute to the smoothing effect of DCT-PLS

and therefore reduced noise in the predictions. Fig. 7d is for a location near the equator where ESA CCI SM is normally permanently masked due to the presence of dense vegetation (i.e., 100 % filled values). The example time series shows how the additional C- and L-band retrievals in this region form a consistent although somewhat noisy record, which matches reasonably well with the in situ data (anomaly correlation of 0.54). Fig. 7e is located in a complex topography on the Tibetan plateau, with significant amounts of missing data points due to soil freezing. The performance of ESA CCI SM is generally poor in

this region, and the gap-filled product shows only little variation and often contradicts the (sparse) in situ measurements. The available observations are very noisy in this location, and hence do not form a consistent time series with the gap-filled data points.

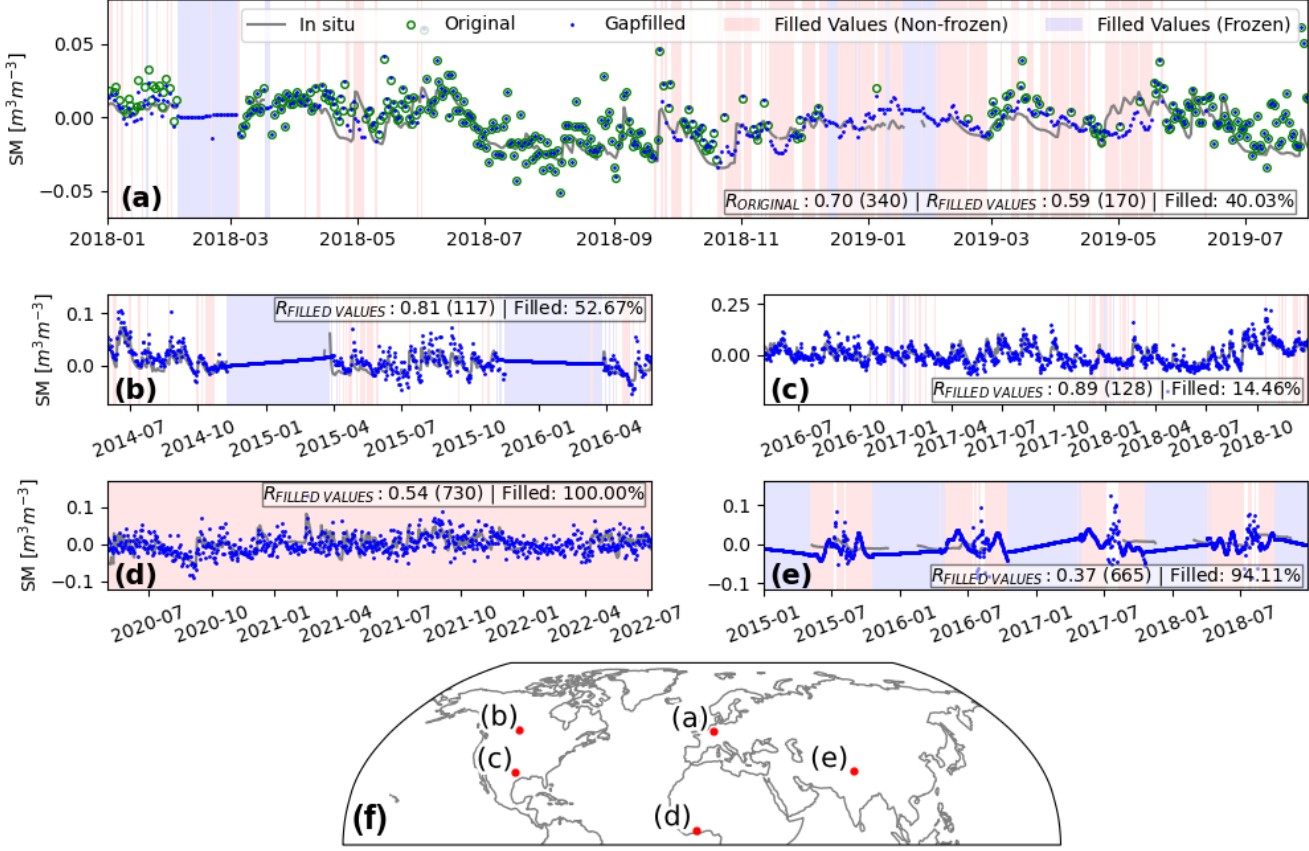

**Figure 7.** Selection of time series from the original and gap-filled ESA CCI SM data set and available in situ measurements. The locations of (a)-(e) are indicated in (f). All statistics are based only on data from the shown sub-periods. R-scores are verified for $p < 0.05$ and based on the number of data points given in brackets. For more information about the chosen locations, see Appendix Table A1.

### 6.2.2 Global anomalies and uncertainties

Global monthly anomalies and uncertainties over time are shown in Fig. 8. The similarity in anomalies before (Fig. 8a) and after
(Fig. 8c) gap-filling indicates that large-scale deviations from normal conditions are equally well represented in the gap-filled
as in the original data. The main differences (Fig. 8e) are found in equatorial regions (mainly due to the additional observations
used) and in transition zones/periods between frozen and unfrozen soil moisture.

Uncertainties are high at the start of the record, when data coverage is lowest (compare Fig. 8b,d to Fig. 1b), and for equato-
rial latitudes due to reduced coverage and observation quality. Uncertainties of the gap-filled values often exceed 0.1 $m^3m^{-3}$
in areas where ESA CCI SM measurements are seasonally masked due to frozen soils, and linear interpolation potentially not





sufficiently accounts for diurnal freeze-thaw processes, evaporation and other factors. However, it is important to note that these higher uncertainties reflect the inclusion of additional data points rather than a decline in dataset quality.

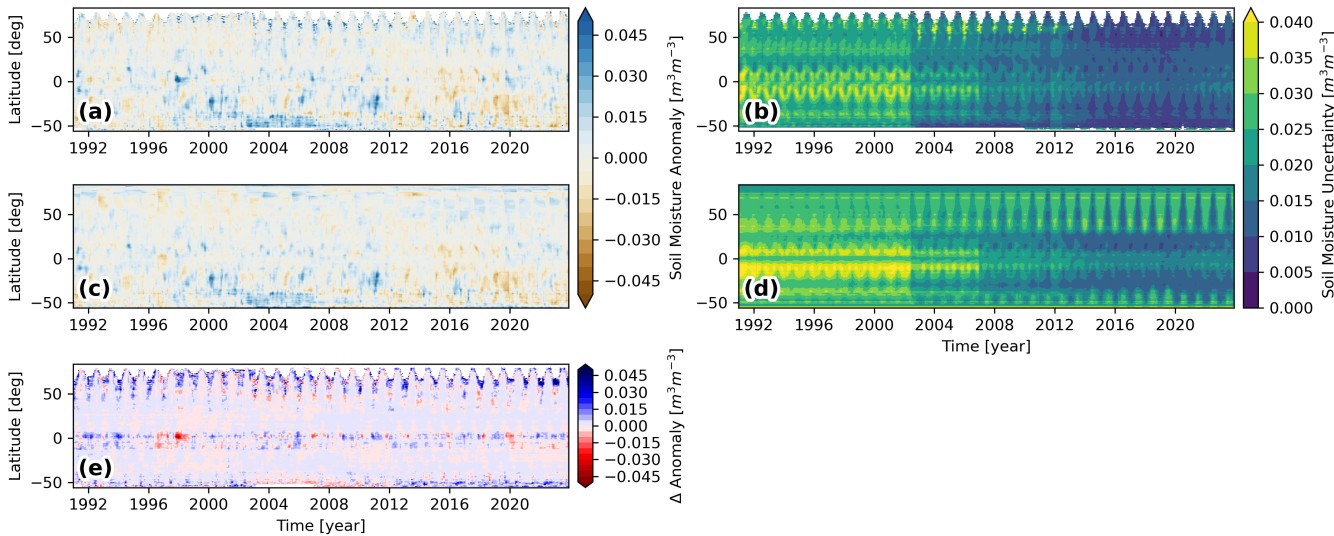

**Figure 8.** Time-latitude diagrams of monthly soil moisture anomalies and soil moisture uncertainties as in the original COMBINED (a), (b) and *GAPFILLED* (c), (d) product. Differences between (a) and (c) are shown in (e).

### 6.2.3 Zonal anomalies and trends

Fig. 9 shows annual anomalies expressed as Z-scores (Vreugdenhil et al., 2022) for different latitude bands of the original observation-only ESA CCI dataset, its gap-filled counterpart, and two reanalysis products. Gap-filling can influence annual anomaly estimates, particularly when large portions of observational data are (systematically) masked. This effect is evident in the global estimates (Fig. 9a), where the 1991-2023 trend shifts from negative to neutral after gap-filling. The most significant discrepancies between the original and gap-filled ESA CCI SM data occur during the first third of the time series (1992–2002), a period with limited observation coverage, compared to the era following the launch of AMSR-E in 2002. The gap-filled data are drier than the original and the reanalysis products. From 2002 to 2016, we find good agreement between all four datasets, until the two CCI (and reanalysis) records start to diverge again, with gap-filled anomalies exceeding those of the original data. The extreme dry anomaly in the original dataset in 2019 is mitigated after gap-filling, resulting in a closer alignment with reanalysis. During the last four years, CCI and GLDAS agree well, while ERA5-Land reached the lowest value on record in 2023. For the Southern Hemisphere (Fig. 9b), we find only minor differences between the original and gap-filled data, mainly in the beginning and at the end of the time series. We attribute this partly to the good initial data coverage, but it also indicates that the integration of additional observations for the previously permanently masked tropical rainforest regions was successful. The gap-filled time series is now slightly less variable and closer to GLDAS-Noah. The overall trend for the Southern Hemisphere has slightly changed, from neutral to negative, in line with the reanalysis products. Looking at Z-scores








for the tropical zone (Fig. 9c) separately, we find a negative soil moisture trend after gap-filling. This trend, as well as the

annual anomalies in general, align better with the reanalysis products, compared to the original data. Some differences are found during the first five and the last 10 years, when gap-filling has better aligned CCI with GLDAS. Fig. 9d comprises the data from a wide range of climatic regions, from arid over temperate to boreal conditions. This subset contains the majority of points over land for the Northern Hemisphere (excl. Greenland). From 1991 to 2003 we find drier, from 2013 to 2020 wetter conditions after gap-filling, which matches with the observations from 9a in the same period, where it had a similar effect

on the long-term trend in the data. The largest discrepancies between the original and gap-filled data are found for the boreal zone (Fig. 9e). The effect of the interpolation of seasonally missing values is clearly visible and has led to a switch from a slight negative to a strong positive trend, which is in contrast to ERA5-Land. However, it should be noted that the reliability of reanalyses in this case is also unknown. This is also evident from the discrepancies between GLDAS Noah and ERA5-Land in this zone.

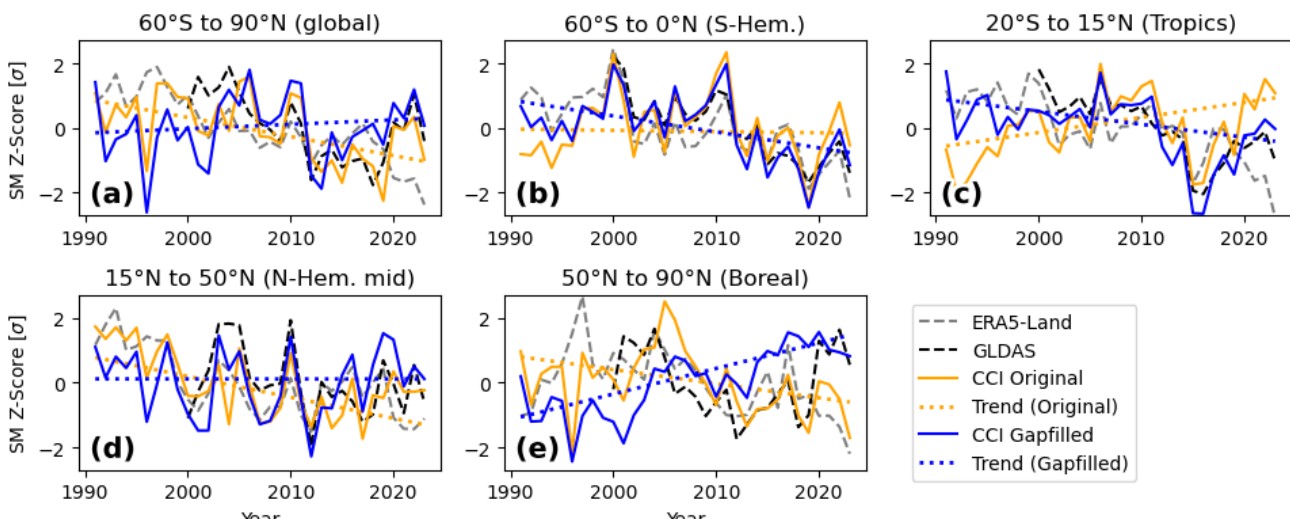

**Figure 9.** Annual soil moisture anomaly Z-scores across different latitude bands: (a) global, (b) Southern Hemisphere, (c) (sub)tropical zone, (d) arid/temperate zone, (e) boreal zone. The dotted coloured lines indicate the zonal trends in the original and gap-filled data.

We further assess global trend changes in Fig. 10, showing Theil-Sen slopes computed from annual averages between 1991 and 2023, with a Mann-Kendall test for statistical significance (Dorigo et al., 2012). Fig. 10a is for the original ESA CCI SM data (with gaps), and 10b the same for the gap-filled product. 10c summarizes the differences in terms of change in trend direction and significance. Only a very small number of initially detected significant trends changed their direction due to gap-filling, either from positive to negative (1.5 %) or vice versa (1.4 %). Affected regions are in the NW United States, Siberia and

Myanmar, where positive trends are found after gap-filling, as well as Central Africa and some spots in South America, where the trend direction was inverted to negative. More prominently, large parts of the Northern Hemisphere, where previously no significant increase in soil moisture was found, now show a positive trend. This is the case for large parts of Russia, Alaska, the





NW United States, and Canada. The opposite is often found in the Southern Hemisphere, mainly in Central and Western Africa and South America, but also in NE China and Mongolia. The previously masked regions covered by tropical rainforest do not
show spatially consistent trends after gap-filling. Especially for parts of Brazil and Columbia, trends vary spatially, while a consistent negative trend is found for the lower half of South America, which only changed in some spots in the far South due to gap-filling.

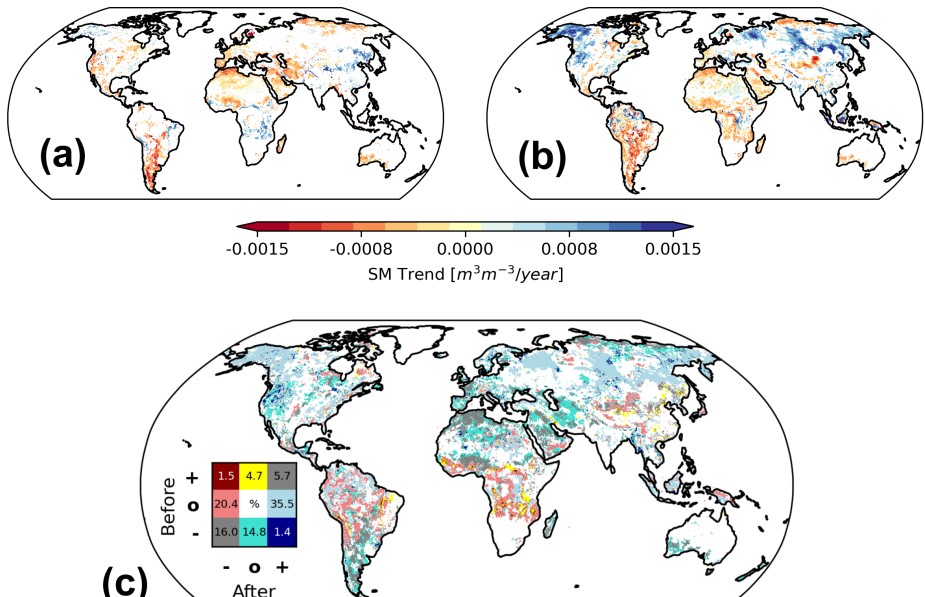

**Figure 10.** Significant ($p < 0.05$) long-term (1991-2023) soil moisture trends before (a) and after (b) gap-filling. (c) shows the change in trend direction and significance: "+" indicates wetting, "-" drying, and "0" non-significant trends before and after gap-filling, respectively. Inset numbers (%) are relative to the total number of points with a significant trend.

### 6.3   Evaluation of $\sigma_{gapfilling}$

Fig. 11 shows Pearson's R (masked for $p < 0.05$) and ubRMSD, between original daily soil moisture from GLDAS-Noah and
restored GLDAS-Noah points for the imposed ESA CCI SM gaps. The gap-filling algorithm manages to restore the data well (global median $R = 0.81$, $ubRMSD = 2.69\,\mathrm{kg\,m^{-2}}$, corresponding to ~10 % relative error). When data from gaps classified as "frozen" (Sect. 3.2) are excluded, i.e., without values filled by linear interpolation, performance metrics improve (global median $R = 0.89$, $ubRMSD = 2.24\,\mathrm{kg\,m^{-2}}$). By design, regions with low R and high ubRMSD in this analysis match with regions where our uncertainty models predict a lower gap-filling accuracy (Fig. 6d, 8d).





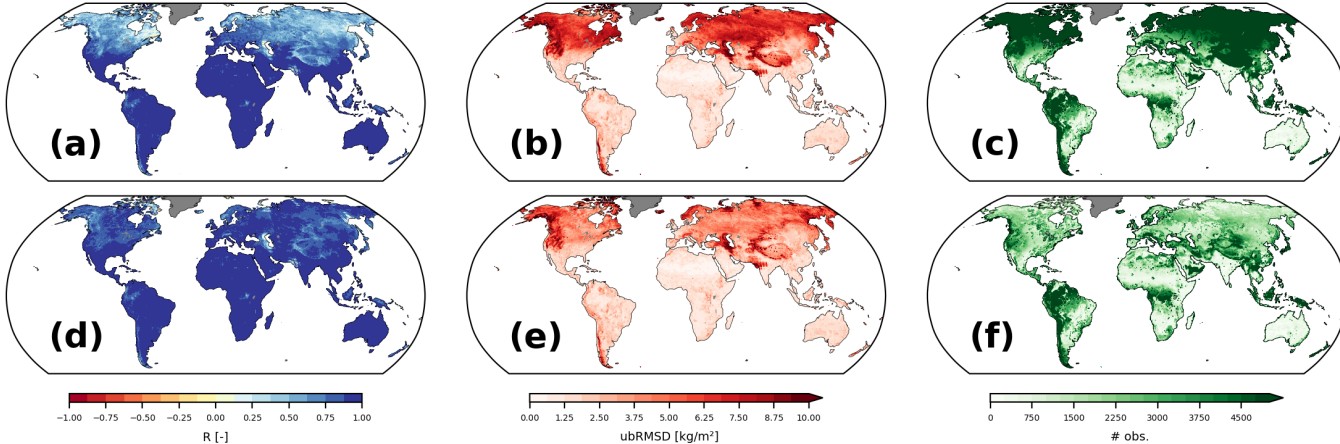

**Figure 11.** Agreement between original and restored GLDAS-Noah surface soil moisture, obtained after imposing data gaps from ESA CCI SM and subsequent filling. The top row is based on all restored data points, the bottom is with frozen periods excluded. Pearson's R in (a) and (d) is masked for $p < 0.05$. ubRMSD is shown in (b) and (e), and the respective sample sizes over the period 2000-2023 in (c) and (f).

### 6.4 *GAPFILLED* soil moisture validation with in situ data

Fig. 12 shows aggregated performance metrics (R and ubRMSD) between FRM in situ sites and the original ESA CCI SM COMBINED product, the new *GAPFILLED* product (original data with filled values), and the subset consisting only of the filled values. All three datasets generally perform on a similar level. In terms of absolute value correlations, the *GAPFILLED* product slightly outperforms the original data. Note, however, that the temporal and spatial coverage differs among the three products. Logically, there is no overlap in observations between the original and the filled values. Fill values typically comprise challenging cases, where satellite retrievals were not possible in the first place. Consequently, it is unsurprising that the performance of only the filled values slightly lags behind the original dataset in terms of absolute values R (Fig. 12a) and ubRMSD (12c), especially for stations near or under dense vegetation ("Tree Cover"). Metrics are slightly more spread for the filled values. A good performance of the *GAPFILLED* product is also found in terms of anomaly R and ubRMSD, indicating that the gap-filling algorithm manages to capture not only the seasonality but also short-term events, on a par with the observations (based on the same 1991-2023 reference period as in Sect. 6.2.1). The station count for anomaly metrics can be lower than for absolute values, as QA4SM excludes time series for which a reliable anomaly cannot be computed (e.g., due to insufficient temporal coverage).



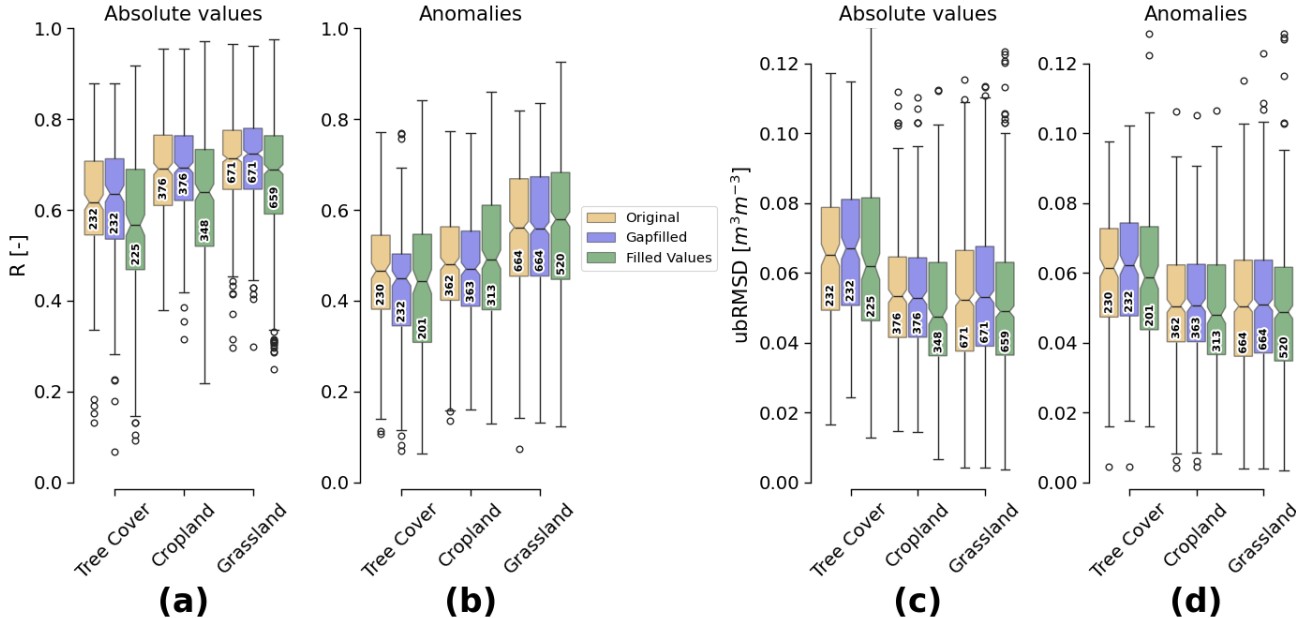

**Figure 12.** Performance metrics between ISMN soil moisture time series and the "original" ESA CCI SM v09.1 COMBINED product, the *GAPFILLED* product, and only the "filled values" of the *GAPFILLED* product. Pearson's R ($p < 0.05$) based on absolute values (a) and anomalies (b), and ubRMSD (c), (d), respectively. The numbers in each box indicate the count of contributing time series.

In Fig. 13, we assess the performance of the filled values separately in "small" and "medium/large" gaps with respect to the
same in situ reference measurements. 13a shows, that the temporal agreement between in situ and filled values decreases for
larger gaps. However, it is less distinct in terms of ubRMSD (Fig. 13b), which barely increases for larger gaps. Note, however,
that there is a severe lack of reference data for the larger gaps, mainly due to the unavailability of in situ measurements during
winter. Hence, it was not possible to evaluate "medium" and "large" gaps separately.

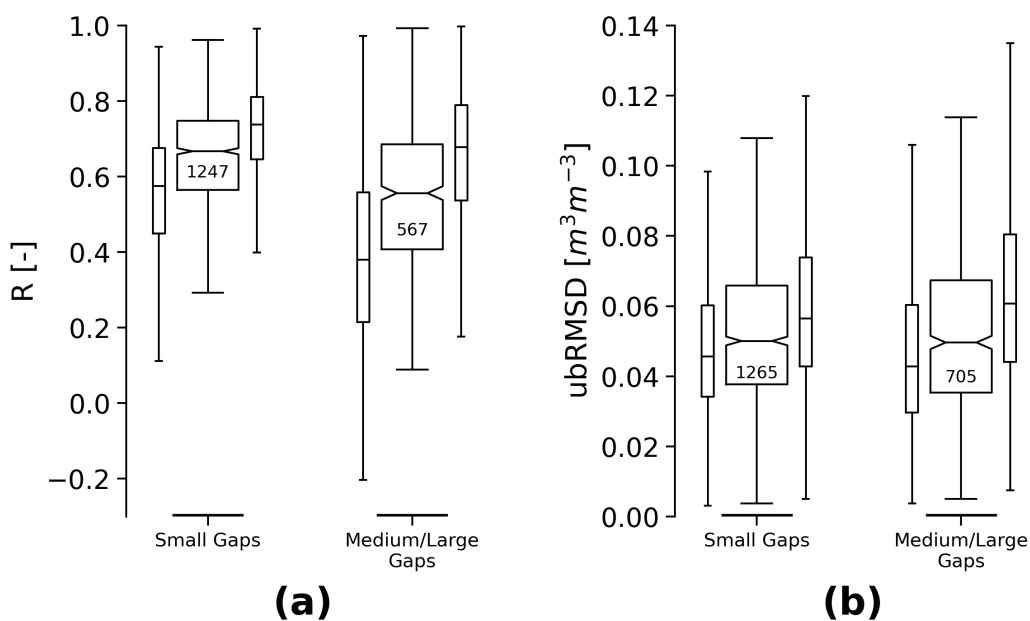

**Figure 13.** Performance metrics between ISMN soil moisture and only the filled values in the *GAPFILLED* product, separated by their Euclidean distance (ED) to the nearest observation: Small Gaps ($ED < 2$), and Medium/Large Gaps ($ED \geq 2$). The shown metrics are (a) Pearson's R (only $p < 0.05$), and (b) ubRMSD. Narrow boxes indicate the upper and lower boundary values of the 95 % confidence interval for each data point. The numbers in each box refer to the sample size (number of time series)

## 7 Code and data availability

The ESA CCI SM v09.1 *GAPFILLED* dataset is available at https://doi.org/10.48436/s5j4q-rpd32 (Preimesberger et al., 2024). It contains the gap-filled original soil moisture observations, gap-free uncertainty estimates, as well as the pure predictions before reimposing the observations (i.e., for missing data points as well as in place of the original measurements). We also provide two binary masks in the data to separate (original) observations and filled values in the gap-filled soil moisture field, and to differentiate between DCT-PLS predictions and linear interpolation values over frozen periods.

Our python implementation of the DCT-PLS algorithm is based on the original Matlab implementation by Garcia (2010), and provided as part of the pytesmo soil moisture toolbox package (Paulik et al., 2024) at https://github.com/TUW-GEO/pytesmo.

## 8 Discussion and conclusions

The gap-filling framework around the DCT-PLS algorithm effectively estimates soil moisture for data gaps in the ESA CCI SM dataset. Choosing a univariate algorithm ensures that the resulting product relies solely on soil moisture observations, 440 independent of external satellite or model datasets. Additionally, DCT-PLS is a performant algorithm suitable for operational data production, as envisaged for the Copernicus Climate Change Service (C3S) (Dorigo et al., 2024b).



We incorporated additional soil moisture retrievals from L- and C-band in equatorial latitudes with dense vegetation cover as anchor measurements for the interpolation algorithm. Although these measurements are often noisy and therefore justifiably excluded from the original ESA CCI SM data, they served as viable support points for gap-filling, leading to a moderate
agreement with available in situ measurements in the area.

Linear interpolation during periods when soil moisture is frozen may appear simplistic, but follows the logic that there is no loss of water from the soil resulting in short-term changes. However, our method does not account for potential sub-pixel freeze/thaw dynamics. Further research is needed to better understand soil moisture dynamics at coarse scales in such cases, enabling satellite retrievals and improved gap-filling predictions in the future. In the final data files, we provide a separate flag
to identify cases where linear interpolation was applied, so that users can filter them out or replace them if necessary.

To provide users with a quality estimate for the gap-filled values, we developed a method to quantify the uncertainties in our predictions. They are combined with the observation uncertainties and included as a separate field in the dataset. These uncertainties are generally higher for larger gaps and regions with dense vegetation compared to areas with better initial coverage. We used 3-dimensional Euclidean distances to the nearest valid data point as an estimate of gap size, but in practice,
it could matter whether this point is close temporally or spatially, due to different levels of soil moisture autocorrelation in these dimensions (Piles et al., 2022). Evaluating these uncertainties remains challenging due to limited reference data. While correlation with in situ measurements decreases for larger gaps, this could not be confirmed in terms of ubRMSD. We do not consider the current uncertainty estimates final, and encourage further research on this topic. Methods such as Gaussian process regression (Gelfand and Schliep, 2016) have been used to fill measurement gaps in time series data in the past (De Caro et al.,
2023). They provide uncertainty estimates based on the variance of multiple predictions, but for the same reason were also found to be computationally too demanding (Heaton et al., 2019) for operational application in a global record such as ESA CCI SM and C3S.

We observed that the *GAPFILLED* product performs similarly to the original observations for the absolute values (including the climatological signals) and short-term fluctuations (anomalies). In some cases, the filled values match better with indepen-
dent reference data than the original observations, likely due to the smoothing effect of DCT-PLS. Similar outcomes have been observed with other smoothing algorithms, such as the exponential filter model, that is used to estimate root-zone soil moisture from surface measurements (Pasik et al., 2023; Wagner et al., 1999; Albergel et al., 2008). These conclusions are based on evaluations using in situ data and from successfully restoring artificially introduced gaps in the gap-free GLDAS-Noah dataset. One downside of the latter approach is, that model and satellite soil moisture can differ, for example, in terms of noise level,
the representation of extreme events, or autocorrelation. This can be due to limited model representations, for example due to disregarded irrigation signals (Zaussinger et al., 2019) or unaccounted latent water influx from rivers (van der Schalie et al., 2022). Thus, conclusions drawn from the restoration of model-based data may not fully apply to satellite observations in some regions, but still give valuable insights. Regional parameterization of the DCT-PLS algorithm resulted in spatial variations in $s$. The use of GCV-score optimisation for $s$ (Garcia, 2010) follows the recommendation of Wang et al. (2012). While the
impact on the predictive skill compared to using a single global $s$-value was not extensively evaluated, we found a good global performance in restoring artificially excluded data.



The *GAPFILLED* dataset is suitable for both long-term and event-based studies. Our comparisons, as well as other studies, have shown that derived statistics (such as changes in annual anomalies) can be affected by gap-filling (Bessenbacher et al., 2023). However, in complex environments with little or no input observations, such as mountainous regions, minimal variability is observed in our data. Similarly, using a smoothing algorithm such as DCT-PLS for interpolation may be detrimental in terms of preserving small-scale extreme events. For applications that do not require a pure soil moisture based dataset, multivariate methods might therefore still be preferred. These studies could potentially use our data as starting values for further adjustment.

# 9 Appendix

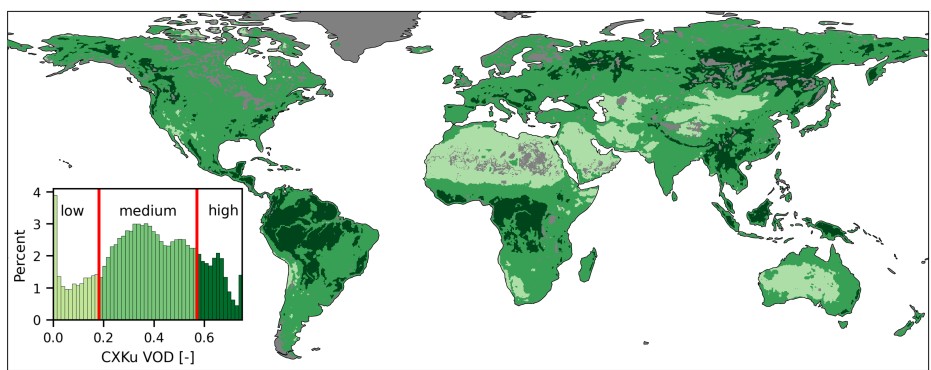

**Figure A1.** VOD classification based on VODCA v2 CXKu multiband 2000-2023 average conditions.

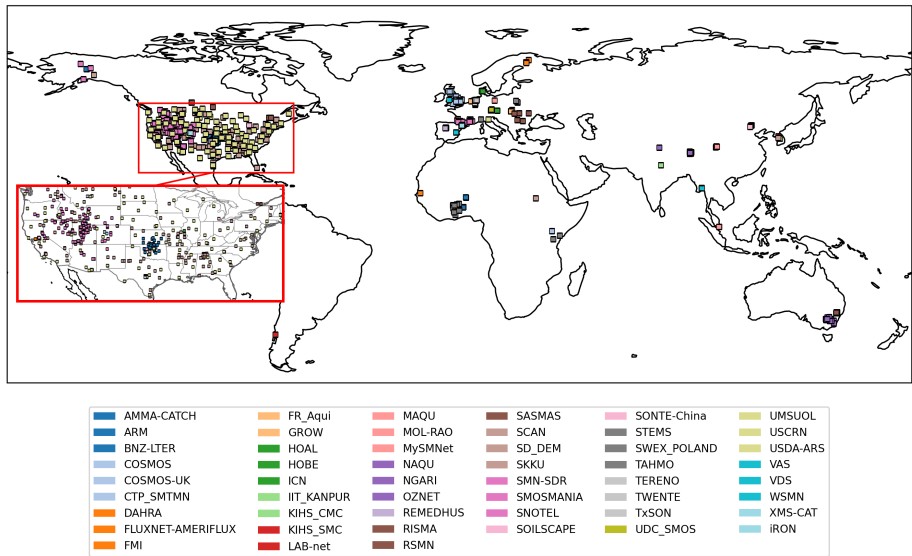

**Figure A2.** ISMN Fiducial Reference Measurements (0-0.1 m depth) in ISMN v20240314 on qa4sm.eu (network references in Appendix Table A2).





**Table A1.** ISMN sensor locations and properties from Fig. 7.

|  | Network Name | Station Name | Sensor Type | Latitude [deg] | Longitude [deg] | Elevation [m] | Depth [cm] | Landcover Class | Climate Class |
|---|---|---|---|---|---|---|---|---|---|
| (a) | TERENO | Schoeneseiffen | Hydraprobe-II-Sdi-12 | 50.5149 | 6.37559 | 612 | 5 | 130 | Cfb |
| (b) | RISMA | SK2 | Hydraprobe-II-Sdi-12 | 51.33504 | -106.5639 | 590 | 0-5 | 130 | Dfb |
| (c) | SCAN | Stephenville | Hydraprobe-Digital-Sdi-12-(2.5-Volt) | 32.25 | -98.2 | 407.896 | 5.08 | 130 | Cfa |
| (d) | TAHMO | BiaSHTSDebiso | TEROS12 | 6.66462 | -3.09725 | 228.0 | 10 | 130 | Aw |
| (e) | NGARI | SQ08 | 5TM | 32.55603 | 79.84004 | 4308.0 | 5 | 200 | Dsb |

**Table A2:** Soil Moisture FRM Networks available through QA4SM

| Network | Reference |
|---|---|
| AMMA-CATCH | Pellarin et al. (2009); Mougin et al. (2009); Cappelaere et al. (2009); Rosnay et al. (2009); Lebel et al. (2009); Galle et al. (2015) |
| ARM | Cook (2016, 2018) |
| COSMOS | Zreda et al. (2008, 2012) |
| CTP_SMTMN | Yang et al. (2013) |
| DAHRA | Tagesson et al. (2014) |
| FLUXNET-AMERIFLUX | - |
| FMI | Ikonen et al. (2018, 2016) |
| FR_Aqui | Al-Yaari et al. (2018); Wigneron et al. (2018) |
| GROW | Xaver et al. (2020); Zappa et al. (2019, 2020) |
| HOAL | Blöschl et al. (2016); Vreugdenhil et al. (2013) |
| HOBE | Bircher et al. (2012); Jensen and Refsgaard (2018) |
| ICN | Hollinger and Isard (1994) |
| IMA_CAN1 | Biddoccu et al. (2016); Capello et al. (2019); Raffelli et al. (2017) |
| IPE | Alday et al. (2020) |
| KIHS_CMC | - |
| KIHS_SMC | - |
| LAB-net | Mattar et al. (2016, 2014) |
| MAQU | Su et al. (2011); Dente et al. (2012) |
| MOL-RAO | Beyrich and Adam (2007) |
| NAQU | Su et al. (2011); Dente et al. (2012) |





| Network | Reference |
| --- | --- |
| NGARI | Su et al. (2011); Dente et al. (2012) |
| OZNET | Smith et al. (2012); Young et al. (2008) |
| PTSMN | Hajdu et al. (2019) |
| REMEDHUS | González-Zamora et al. (2019) |
| RISMA | Ojo et al. (2015); J. (2011); Canisius (2011) |
| RSMN | - |
| SASMAS | Rüdiger et al. (2007) |
| SCAN | Schaefer et al. (2007) |
| SD_DEM | Ardö (2013) |
| SMN-SDR | Zhao et al. (2020); Zheng et al. (2022) |
| SMOSMANIA | Calvet et al. (2016); Albergel et al. (2008); Calvet et al. (2007) |
| SNOTEL | Leavesley et al. (2008); Leavesley (2010) |
| SOILSCAPE | Moghaddam et al. (2011); MOGHADDAM et al. (2016); Shuman et al. (2010) |
| SWEX_POLAND | Marczewski et al. (2010) |
| TAHMO | - |
| TERENO | Zacharias et al. (2011); Bogena et al. (2018, 2012); Bogena (2016) |
| UDC_SMOS | Schlenz et al. (2012); Loew et al. (2009) |
| UMBRIA | Brocca et al. (2011, 2008, 2009) |
| UMSUOL | - |
| USCRN | Bell et al. (2013) |
| VAS | - |
| VDS | - |
| iRON | Osenga et al. (2021, 2019) |

*Author contributions.* WP, PS, WD designed the study. WP performed the analysis and wrote the paper. All authors contributed to discussions about the methods and results and provided feedback on the paper and data.

*Competing interests.* The contact author has declared that none of the authors has any competing interests.



*Acknowledgements.* This dataset was produced with funding from the European Space Agency (ESA) Climate Change Initiative (CCI) Plus Soil Moisture Project (CCN 3 to ESRIN Contract No: 4000126684/19/I-NB "ESA CCI+ Phase 1 New R&D on CCI ECVS Soil Moisture").

The initial dataset was designed under the Horizon 2020 Global Gravity-based Groundwater Product (G3P) Project supported by the European Commission (grant no. 870353). Operational implementation is supported by the Copernicus Climate Change Service implemented by ECMWF through C3S2 312a/313c.



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
