# Peer review of "ESA CCI Soil Moisture GAPFILLED: An independent global gap-free satellite climate data record with uncertainty estimates"

_Earth System Science Data, 2024_

## Author Comment (AC1)

[Figure]

Fig. R1 - 1991-2023 average uncertainty estimates (a) in satellite observations (TCA) used for the GAPFILLED product, (b) only the filled values in the GAPFILLED product, (c) the final GAPFILLED product overall.

[Figure]

Fig. R2 - Soil moisture climatologies for a region in the Amazon rainforest and Pearson R between them, for GLDAS Noah, ASCAT and SMAP.

[Figure]

*Fig. R3 - The impact of below or close-to freezing temperature on soil moisture signals which motivate the use of 30-day mean values for linear interpolation (red lines). (a) Time series from the HOBE network in Denmark; (b) Same for the FMI network (Finland); "x" indicate flagged / excluded measurements, which are not used for the 30d average.*

---

## Author Comment (AC2)

[Figure]

Fig. R1 - The impact of below or close-to freezing temperature on soil moisture signals which motivate the use of 30-day mean values for linear interpolation (red lines). (a) Time series from the HOBE network in Denmark; (b) Same for the FMI network (Finland); "x" indicate flagged / excluded measurements, which are not used for the 30d average.

[Figure]

Fig. R2 - Proposed update to time series plots from manuscript Fig. 17, with the green circles (and additional stats) included in all subplots compared to the original figure.

---

## Author Response (AR1)

**Note: The following replies are the same as provided for the Discussion at https://essd.copernicus.org/preprints/essd-2024-610/. In addition, text changes are now included and marked in RED (please also the see uploaded difference PDF file).**

**Response to Reviewer #1**

We thank the reviewer for their constructive comments. We address them separately below.

1) **Fig. 6 The highest uncertainty in the gap-filled product is in the African subtropical zone, whereas in the tropical zone in Africa, and even more so in the Amazon, the uncertainty is lower. Given the uncertainty in the retrievals associated with dense vegetation of the tropical region, what would be the reason for the final uncertainty to be lower than in the subtropics?**

We agree that uncertainties in the highlighted parts of the tropical zone are overall likely underestimated (Fig. R1a).

Uncertainties of the GAPFILLED data depend on:
(1) the gap sizes: our gapfilling uncertainty predictions are based on (empirical-statistical) models, for which both denser vegetation and larger gaps generally lead to increased uncertainties.
(2) (random) uncertainties of the retrievals for available data points: these are estimated from Triple Collocation Analysis (TCA) random error estimates, and propagated to the gaps.

Ad (1): With the added "anchor" observations over tropical rainforests (p. 24, line 443), the (3-dimensional) gap sizes are reduced significantly, leading to similar gap distributions as in subtropical zones. This means that the used uncertainty models (manuscript Fig. 4b) yield similar results in areas with moderately large gaps. The observed differences, however, don't affect all "high VOD" areas from our classification, but only a subregion. While an additional class for "very dense" vegetation could somewhat correct for this, we think our models are not the driving factor, as the relative difference between the "high" and "medium" VOD model is smaller than what is observed.

Ad (2): This is a more likely cause, as over rainforests the TCA seems to underestimate random errors. The averaged TCA uncertainties shown in Fig. R1 below (for the period 1991-2023) indicate that the overall TCA uncertainty is lower in some areas of the tropical zone than in the subtropics. This originates from the lower uncertainties of the observations in these areas (Fig. R1a), which are then propagated to the gapfilled values (manuscript Eq. 10), where overall the uncertainties increase (Fig. R1b), but are still too low in the final product (Fig. R1c). This is also confirmed when looking at differences in the soil moisture signals (Fig. R2), in this case the climatologies over a subset of the affected area. We find a lower correlation between GLDAS Noah and the satellite products, while ASCAT and SMAP in this example agree better. This explains the lower random error levels for the satellite data, as the errors are essentially attributed to GLDAS Noah.

We conclude that the underlying reasons are too low uncertainty estimates found for the observation data, which probably do not sufficiently represent the challenging retrieval of soil moisture under a dense vegetation layer (even for L- and C-band data).

However, we consider that improving observation uncertainty characterisation and/or improving soil moisture retrieval under dense vegetation layers are both outside the scope of this paper. We will, however, add a paragraph based on the analysis above, to discuss this topic in the manuscript.

**The following text was added to chapter 6.2 (L326 ff)**

The uncertainties are within the range of 0 to 0.1 m$^3$ m$^{-3}$, which is expected from the defined models for $\sigma_{gapfilling}$ and the value range of the original (triple collocation) error estimates $\sigma$observations (Gruber et al., 2016, 2019). [...]

While uncertainties increase as expected in most regions covered by dense vegetation, an inverse effect appears in some areas of the northwestern Amazon and the central African rainforest (Appendix Fig. A2), where the uncertainties of the *GAPFILLED* product are lower than in parts of the surrounding subtropical zones. Our analysis of this phenomenon shows that – although uncertainties in the *GAPFILLED* product increase according to $\sigma_{gapfilling}$ – the initial uncertainty estimates based on triple collocation analysis $\sigma_{observations}$ in the affected regions are too low (Fig. A2a). This likely results from the poor agreement between satellite and model data in these areas, whereas similar soil moisture climatologies are found in the (independent) active and passive satellite products (Fig. A2d).

**The following text was added to chapter 8 (L459 ff)**

However, we found that the retrieval uncertainty estimates based on triple collocation analysis for these areas are often too low, which can also affect the (propagated) uncertainties of the *GAPFILLED* product. We conclude that not only the retrieval of soil moisture under very dense vegetation layers remains challenging, but also the estimation of associated uncertainties warrants further research. Studies utilising these retrieval uncertainties, including ours, could apply them to provide better estimates of the quality of derived products based on these data.

**The following figure was added to Appendix A**

[Figure]

**Figure A2.** 1991-2023 average uncertainty estimates in parts of the tropical climate zone of (a) satellite observations used for the *GAP-FILLED* product (based on triple collocation analysis), (b) only the filled (propagated) values in the *GAPFILLED* product, (c) the final *GAPFILLED* product overall (circles indicate areas with too low uncertainty estimates). (d) shows the mean soil moisture climatologies of SMAP, ASCAT and GLDAS Noah over the area outlined by the red bounding box in (a).

**2) Fig 7. Negative values of SM are present for all locations. I assume that is the result of applying the scaling algorithm. So, I assume both in-situ and the gap-filled datasets were scaled. From the text, it was not explicitly clear. I suggest adjusting the text to reflect scaling applied to both datasets as well as modifying the Y-axis labels in Fig. 7, highlighting that the values do not represent the absolute value of SM in m3/m3 but a scaled value.**

Manuscript Fig. 7 shows soil moisture anomalies (relative to the long-term 1991-2023 average), where values below 0 refer to "drier-than-usual" conditions. However, we realize that this is not sufficiently explained in the figure caption, and not reflected by the y-axis labels / figure title. We will change the descriptions of Fig. 7 in the manuscript accordingly.

**Fig. 7 and caption were changed as follows (left=before, right=after)**

[Figure]

**Figure 7.** Selection of time series from the original and gap-filled ESA CCI SM data set and available in situ measurements. The locations of (a)-(e) are indicated in (f). All statistics are based only on data from the shown sub-periods. R-scores are verified for $p < 0.05$ and based on the number of data points given in brackets. For more information about the chosen locations, see Appendix Table A1.

**Figure 7.** Selection of soil moisture anomaly time series from the original and gap-filled ESA CCI SM data set and available in situ measurements (relative to the 1991-2023 climatological average). The locations of (a)-(e) are indicated in (f). All statistics are based only on data from the shown sub-periods. R-scores shown in bold are statistically significant ($p < 0.05$) and are based on the number of data points indicated in parentheses. For more information about the chosen locations, see Appendix Table A1.

**3) In Fig 7 (a) there is one gap-filled value associated with a non-frozen state in the middle of the frozen state. The difference between the linearly interpolated values for the freeze-to-thaw period and that one value in the middle seems significant. Would that be a corner case not covered by the combination of carrying over the last value before the ground is frozen and interpolating between the last value before freezing and the first value after thawing (with the 30-day mean)?**

The highlighted data point is indeed an example of inconsistent predictions between the two applied interpolation algorithms. We "overwrite" DCT-PLS predictions in periods of negative ERA5 soil temperature with values from linear interpolation.

It is, however, as pointed out by the reviewer, an edge case, where i) a single unfrozen data point was found in the middle of a prolonged period of negative temperature, which ii) was also not picked up by a satellite (as the surface temperature measured by the satellite was too low for soil moisture retrieval) at the time. Hence, the value comes from the DCT-PLS interpolation algorithm, which is known to produce too low values at the transition between frozen/unfrozen soil moisture (Wang et al., 2012) which was one motivation for us to introduce the linear interpolation in the first place.

A relatively simple fix would be to apply a more strict classification threshold for frozen soils, e.g., by picking a higher temperature threshold value (e.g. 4 °C instead of 0 °C, recommended by Gruber et al., 2020) to apply linear interpolation, or setting a minimum number of consecutive "unfrozen" data points, which would in the example case remove the outlier as linear interpolation would be applied over the whole period.

We will add a note to the discussion of manuscript Fig. 7a, and consider changing the temperature threshold for future data releases. As we provide users with the required information to identify these edge cases (via the binary masks that come with the data), we consider that no reprocessing is required for the version supported by the manuscript.

**The following text was added to chapter 6.2.1 (L347 ff) - related to this, please see also the changes for comment 4**

This plot also shows how combining data from both the DCT-PLS and linear interpolation in a single time series based on soil temperature can lead to inconsistencies for some edge cases (February 2018). In this case, an individual, unobserved "non-frozen" data point is found in the middle of a prolonged frozen period. Differences in the predictions of the two interpolation algorithms can lead to outliers in this case.

4) **In the case of Fig 7 (a), it seems to be due to the fact that the 30-day mean after the thawing is a lot higher than the first value after the thawing, so the linear interpolation seems to actually create more fluctuation in the predictions rather than mitigating them. I would assume SM can change drastically in spring in the first 30 days after thawing. Have you considered other values for this window and the sensitivity of the results to these values?**

We have experimented with using only the first and the last data point before/after a period of frozen soils, as well as different thresholds, but opted for a relatively large window of 30 days because we found that noisy observations or even outliers before/after a period of frozen soils otherwise have a very strong impact on the interpolation outcome (clearly visible in Fig. R3). Consequently, a strong gradient in the linear interpolation over time was found with a smaller window, which is not realistic if we assume that soil moisture content should remain unchanged while frozen in the soil over that period.

The current threshold could be further refined, for example by considering different onset periods for different climates globally, but for now it was found to be the best compromise between a stable interpolation and a smooth transition between interpolation and observation data.

We will update the discussion in the text accordingly, to make this point clearer.

**The following text was added to chapter 8 (L465 ff)  - related to this, please see also the changes for comment 3**

We use the average of observations from 30 days before and after periods when soil moisture is frozen as bounding points for linear interpolation. The threshold was determined based on regional experiments with satellite and in situ data, and was selected as the best compromise between stable interpolation and a smooth transition between interpolated and observed data. However, it could be further refined, for example by considering different onset periods for different climates globally. Our approach has the drawback that it relies on external soil temperature information, which can introduce outliers in the time series when prolonged periods of frozen soils (<0 °C) are interrupted by brief thawing events. This issue could be mitigated by adopting a slightly stricter threshold value (e.g., 4 °C, in line with Gruber et al. (2020)) and/or by setting a minimum number of consecutive days above 0 °C before a data point is classified as "unfrozen".

**References**

Gruber, A., De Lannoy, G., Albergel, C., Al-Yaari, A., Brocca, L., Calvet, J.-C., Colliander, A., Cosh, M., Crow, W., Dorigo, W., Draper, C., Hirschi, M., Kerr, Y., Konings, A., Lahoz, W., McColl, K., Montzka, C., Muñoz-Sabater, J., Peng, J., Reichle, R., Richaume, P., Rüdiger, C., Scanlon, T., van der Schalie, R., Wigneron, J.-P. & Wagner, W. (2020) 'Validation practices for satellite soil moisture retrievals: What are (the) errors?', *Remote Sensing of Environment*, 244, p. 111806. Available at: https://doi.org/10.1016/j.rse.2020.111806

Wang, G., Garcia, D., Liu, Y., de Jeu, R. & Dolman, A.J. (2012) 'A three-dimensional gap filling method for large geophysical datasets: Application to global satellite soil moisture observations', *Environmental Modelling & Software*, 30, pp. 139-142. Available at: https://doi.org/10.1016/j.envsoft.2011.10.015

**Response to Reviewer #2**

We thank the reviewer for their constructive comments. We address them separately below. Supporting figures are attached as "FigR2.pdf"

1) **p5 lines 115-126: ERA5-Land has soil temperature values at the same 0-7cm depth as ERA5. Hence, why use ERA5 soil temperature for determining frozen periods, not ERA5-Land?**

Previous research has shown that the difference in skill between ERA5 and ERA5-Land skin temperature compared to remote sensing estimates is minimal and limited to coastal or topographically complex regions (Muñoz-Sabater et al., 2021). On a global scale, the difference in RMSD with MODIS-based soil temperature is only ~0.2 K (2003-2018 period), and we expect it may become even more difficult to appreciate such differences at the resolution considered for the present study (i.e., after aggregation of ERA5-Land from 0.1 to 0.25 deg). No systematic bias in the global skin temperature predictions between the two reanalysis products is known to the authors. Considering these as main drivers of the onset or fading of frozen periods, we expect no systematic impact on the skill of our proposed approach by exchanging the two products.

2) **Related to this is p11 250-252: How much uncertainty exists in the timing of freeze/thaw transition compared to the 30-day period used to determine interpolation values?**

This depends on various factors which affect the rate at which soil moisture in a satellite pixel freezes. As long as temperatures fluctuate around 0 °C and/or parts of the soil in a satellite pixel transition between frozen/unfrozen state, we expect potential (unflagged) outliers in the measurements, which can affect the interpolation process.
We chose the 30-day threshold based on analyzing the soil moisture signal in a number of affected locations (e.g., Fig. R1). The 30-day period is a compromise that is long enough to reduce the impact of outlier cases, while still being representative of the overall (climatic) conditions (Fig. R1b). Even though the in situ data - compared to the satellite - represent soil moisture for a specific location rather than a larger area, unrealistic (unflagged) fluctuations in the days/weeks before soil moisture is frozen are visible (Fig. R1a). The impact of such outliers is minimized by the use of the 30-day period.

**The following text was added to chapter 6.2.1 (L347 ff)**

"This plot also shows how combining data from both the DCT-PLS and linear interpolation in a single time series based on soil temperature can lead to inconsistencies for some edge cases (February 2018). In this case, an individual, unobserved "non-frozen" data point is found in the middle of a prolonged frozen period. Differences in the predictions of the two interpolation algorithms can lead to outliers in this case."

**The following text was added to chapter 8 (L465 ff)**

"We use the average of observations from 30 days before and after periods when soil moisture is frozen as bounding points for linear interpolation. The threshold was determined based on regional experiments with satellite and in situ data, and was selected as the best compromise between stable interpolation and a smooth transition between interpolated and observed data. However, it could be further refined, for example by considering different onset periods for different climates globally. Our approach has the drawback that it relies on external soil temperature information, which can introduce outliers in the time series when prolonged periods of frozen soils (<0 °C) are interrupted by brief thawing events. This issue could be mitigated by adopting a slightly stricter threshold value (e.g., 4 °C, in line with Gruber et al. (2020)) and/or by setting a minimum number of consecutive days above 0 °C before a data point is classified as "unfrozen"."

3) **p10 Although the method is previously reported, the equations could be more clearly explained for readers that only want to get a brief idea from this paper and not read the original. Does the "3-dimensional case" mean lat-lon-time? What are $n$ (Eq. 4), $n_j$ and $i_j$ (Eq. 5)? Are the ÷ and ∘ element-wise operations? Is $k$ in Eq. 7 iteration index?**

- Yes, 3-dimensional means lat/lon/time in our case.
- In Eq. 4, "n" refers to the number of elements along dimension i of the tensor for which λ is computed. It becomes clear from Eq. 5 (λ is the summand) which shows how λ is computed over all dimensions, where $i_j$ are the values along the dimension and $n_i$ the value count. We propose to change the summand in Eq. 5, to "$λ_j$" instead, and swap Eq. 4 and 5.
- Indeed ÷ and ∘ symbolize the element-by-element division / multiplication (Garcia 2010, Eq 17 & Eq. 18). We will add this information.
- Yes, $ŷ_k$ refers to the realization of $ŷ$ calculated at the $k^{th}$ iteration step.

**The following text was added to chapter 4.1**

- "In the 3-dimensional case (N = 3; time, latitude, longitude) this leads to tensor Λ (Eq. 5)."
- "In fact, a predefined formulation for $λ_i$ of D can be used (Eq. 4), where i are the values along the dimension and n the value count."
- "Together with a realization of the smoothing parameter s, we can define the tensor $Γ^N$ (Eq. 6, ÷ and ∘ symbolize the element-by-element division and multiplication)"
- "k here refers to one realization of $ŷ$, as this step is repeated multiple times when optimising s, as explained later."

**4) p13: Is the VOD classification purely in space or in both space and time?**

Currently, we only use a static map of VOD (space). We will add a sentence to the manuscript to highlight this.
* * *
**The following text was changed in chapter 5**

"To address the latter, we use a static VOD classification map for low, medium, and high VOD based on the 2000-2023 average, derived from the VODCA v2 CXKu archive (compare Appendix Fig. A1)."
* * *
**5) p17 Fig. 7: It is a bit confusing why there are no green circles in panels b-e.**

The idea behind the green circles is to show that original observations are not changed by the gapfilling process and that they form a consistent time series with the predicted value. The circles were excluded from the smaller sub-plots as the information is essentially also provided from the background colors (green circles in subplot (a) correspond to white background color), and to avoid overcrowded subplots. However, as this might be confusing, we propose to add them to all subplots as shown in Fig. R2.

**Fig. 7 was changed as follows (left=before, right=after)**

[Figure]

Figure 7. Selection of time series from the original and gap-filled ESA CCI SM data set and available in situ measurements. The locations of (a)-(e) are indicated in (f). All statistics are based only on data from the shown sub-periods. R-scores are verified for $p < 0.05$ and based on the number of data points given in brackets. For more information about the chosen locations, see Appendix Table A1.

Figure 7. Selection of soil moisture anomaly time series from the original and gap-filled ESA CCI SM data set and available in situ measurements (relative to the 1991-2023 climatological average). The locations of (a)-(e) are indicated in (f). All statistics are based only on data from the shown sub-periods. R-scores shown in bold are statistically significant ($p < 0.05$) and are based on the number of data points indicated in parentheses. For more information about the chosen locations, see Appendix Table A1.

**6) p21 Fig. 11: The mask is said to be masked for p<0.05 but why no apparent masks can be seen?**

Only a few individual pixels are masked by the p-value threshold, e.g. in western India, where the correlation coefficient is not significant. If p<0.05, data points are *retained* (in the original we wrote "masked", which will be changed), which is almost always the case. We will add a note on this and indicate that almost all shown results are statistically significant.

> **Caption of Fig. 11 was changed as follows**
>
> "[...] Pearson's R in (a) and (d) is masked if p > 0.05 (≈ 3 % of all grid cells). [...]"

**7) Is the #obs the summed sizes of the gaps?**

**obs is the number of data points used to compute the statistics, which is indeed the summed size of all gaps in a location in this context. We understand that "number of observations" can be confusing in the gap filling context, and will label the plot "sample size" instead.**

**Figure 11 was changed as follows (before=top, after=bottom)**

before

[Figure]

**Figure 11.** Agreement between original and restored GLDAS-Noah surface soil moisture, obtained after imposing data gaps from ESA CCI SM and subsequent filling. The top row is based on all restored data points, the bottom is with frozen periods excluded. Pearson's R in (a) and (d) is masked for $p < 0.05$. ubRMSD is shown in (b) and (e), and the respective sample sizes over the period 2000-2023 in (c) and (f).

**after**

[Figure]

**Figure 11.** Agreement between original and restored GLDAS-Noah surface soil moisture, obtained after imposing data gaps from ESA CCI SM and subsequent filling. The top row is based on all restored data points, the bottom is with frozen periods excluded. Pearson's R in (a) and (d) is masked if $p > 0.05$ ($\approx$ 3 % of all grid cells). ubRMSD is shown in (b) and (e), and the respective sample sizes (i.e., number of gaps over time) over the period 2000-2023 in (c) and (f).

**8) Can the ubRMSD be converted from kg/m2 to m3 m-3 like elsewhere in the paper?**

Yes, we will convert the units to make them consistent with the rest of the manuscript. The reason for this is that GLDAS soil moisture values are originally provided in kg/m$^2$.

Figure 11 was changed. See previous comment.

**References**

Muñoz-Sabater, J., Dutra, E., Agustí-Panareda, A., Albergel, C., Arduini, G., Balsamo, G., Boussetta, S., Choulga, M., Harrigan, S., Hersbach, H., Martens, B., Miralles, D. G., Piles, M., Rodríguez-Fernández, N. J., Zsoter, E., Buontempo, C., and Thépaut, J.-N.: ERA5-Land: a state-of-the-art global reanalysis dataset for land applications, Earth Syst. Sci. Data, 13, 4349–4383, https://doi.org/10.5194/essd-13-4349-2021, 2021.